# Depth-Breadth Synergy in RLVR:
# Unlocking LLM Reasoning Gains with Adaptive Exploration

**Zhicheng Yang**[1]  **Zhijiang Guo**[1,2]  **Yinya Huang**[3]  **Yongxin Wang**[4]  **Dongchun Xie**[5]  **Hanhui Li**[5]  **Yiwei Wang**[6]
**Xiaodan Liang**[4,5]  **Jing Tang**[1,2]

yangzhch6@gmail

*Project Repo*: **https://github.com/yangzhch6/DARS**

## Abstract

Reinforcement Learning with Verifiable Reward (RLVR) is a powerful paradigm for enhancing the reasoning abilities of Large Language Models, yet its potential is constrained by insufficient exploration along two core dimensions: Depth (problem difficulty) and Breadth (training instance quantity). Our analysis of the popular GRPO algorithm reveals a bias that down-weights difficult, low-accuracy problems, which are crucial for improving reasoning skills. To address this, we introduce **D**ifficulty **A**daptive **R**ollout **S**ampling (**DARS**), a method that re-weights difficult problems by using targeted, multi-stage rollouts. DARS increases the number of rollout outcomes for these harder problems according to our proposed re-balancing schedules and leads to consistent gains in *Pass@K*. We discovered that increasing rollout size alone does not improve performance and may actually impair it. In contrast, scaling the batch size to increase breadth via full-batch updates significantly boosted Pass@1 metrics. This improvement stems from higher token-level entropy, ensuring robust exploration and minimized gradient noise. We further present DARS-Breadth, a combined approach that uses DARS with a large breadth of training data. This method demonstrates simultaneous gains in both *Pass@K* and *Pass@1*, demonstrating that depth and breadth serve as orthogonal and complementary dimensions to unleash the potential of RLVR.

## 1. Introduction

The emergence of reasoning-centric Large Language Models (LLMs) exemplified by OpenAI-o1 (Jaech et al., 2024), DeepSeek-R1 (Guo et al., 2025), and Kimi-1.5 (Team et al., 2025), has pushed the frontier of LLM capability, especially for demanding tasks in complex reasoning such as mathematics and programming. Unlike conventional instruction tuning that relies on human-labeled data or RLHF pipelines that demand an auxiliary, well-trained reward model (Ouyang et al., 2022b; Achiam et al., 2023; Grattafiori et al., 2024), this leap is driven by large-scale Reinforcement Learning with Verifiable Rewards (RLVR; Guo et al. 2025; Zeng et al. 2025) for which correctness can be automatically and deterministically checked. The rewards of RLVR are granted solely when a model's output matches the ground-truth mathematical answer or passes all unit tests for code, allowing scalable verification without manual labeling. RLVR is now regarded as a promising path toward self-evolving LLMs, potentially bringing us closer to more powerful intelligence.

However, existing RLVR frameworks inadequately address the interplay between exploration depth (difficulty scaling) and breadth (iteration instance quantity scaling), which leads to insufficient performance gain for both *Pass@1* and *Pass@K*. In this paper, we conduct a systematic analysis of these two under-exploited dimensions in RLVR.

For the dimension of **depth**, our investigation reveals that existing methods of GRPO (Shao et al., 2024) and its variants (Yu et al., 2025; Liu et al., 2025b), while adept at estimating the advantage of a single rollout, are undermined by a distorted cumulative advantage at the group level. This distortion disproportionately allocates attention to instances of medium difficulty, neglecting high-difficulty instances indispensable for complex reasoning, as illustrated in Figure 1. This bias fundamentally limits depth, the hardest problems a model can learn to solve, and constrains *Pass@K* performance. To counteract this depth neglect, we propose Difficulty-Adaptive Rollout Sampling (**DARS**). DARS performs a lightweight first-stage rollout to estimate per-problem accuracies, then allocates additional compute via targeted multi-stage rollouts to low-accuracy problems. By expanding sampling on hard problems, DARS re-weights the cumulative advantage, making it easier for LLMs to learn 'deep' samples and improving *Pass@K* performance.

---

[1]HKUST(GZ) [2]HKUST [3]ETH AI Center, ETH Zurich [4]MBZUAI [5]Sun Yat-Sen University [6]University of California, Merced. Correspondence to: Jing Tang <jingtang@ust.hk>.

*Proceedings of the 43rd International Conference on Machine Learning*, Seoul, South Korea. PMLR 306, 2026. Copyright 2026 by the author(s).

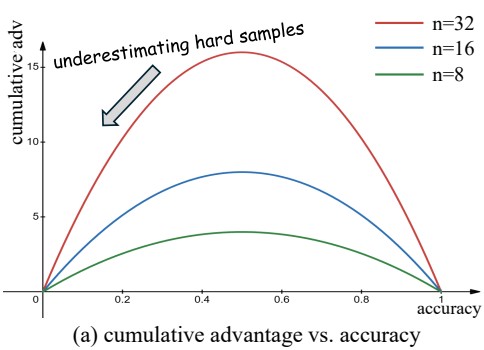

(a) cumulative advantage vs. accuracy

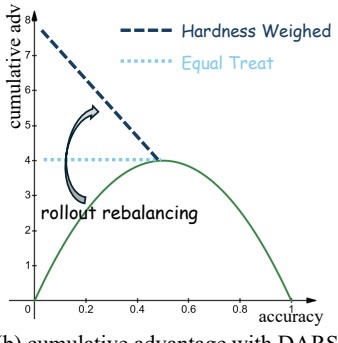

(b) cumulative advantage with DARS

*Figure 1.* Statistical results of cumulative advantage. Group relative advantage calculation methods underestimate high-difficulty problems. $n$ denotes group size.

We further identify **breadth** as the instance quantity consumed in a single iteration. We observe that breadth has a significant impact on the LLM's performance and continuous exploration capability, as shown in Figure 4. In DARS, the dynamic rollout allocation $\Delta n_j$ yields ragged per-question rollout counts that break standard PPO minibatch chunking, so we adopt full-batch updates for multiple PPO epochs as a compatible training design. Based on this, we significantly increase the training batch size. This seemingly simple change dramatically improves *Pass@1* and sustains high token-level entropy throughout training, suggesting that breadth acts as implicit entropy regularization that delays premature convergence. Importantly, the gains from breadth are complementary to those from depth: we present **DARS-Breadth** that combines our DARS with large-breadth training, producing simultaneous boosts in both *Pass@K* and *Pass@1*. Our contributions can be summarized as follows:

- **Cumulative Advantage Bias**: We systematically analyze RLVR depth and breadth, revealing GRPO depth bias: cumulative advantage underweights low-accuracy, high-difficulty samples, capping *Pass@K* performance.

- **Difficulty Adaptive Rollout Sampling**: We introduce DARS to reallocate compute to the hard problems via multi-stage rollout sampling. DARS re-weights the cumulative advantage distribution and quantitatively expands the sparse reward signals for difficult problems. We further show that the ET and HW schedules recover **Log-Odds** and **Maximum Likelihood** objectives, substantially boosting *Pass@K* performance.

- **Depth-Breadth Synergy**: We illustrate that large breadth matters for the *Pass@1* performance in RLVR. By combining DARS with large breadth scaling, we further reveal the complementarity of depth and breadth and acquire simultaneous boosts in both *Pass@K* and *Pass@1* metrics.

## 2. Analyzing RLVR from Depth and Breadth

### 2.1. Depth: The Hardest Problem Sampled in RLVR

We first identify **Depth** as the hardest problem that can be correctly answered in the RLVR training process. In the GRPO training process, groups whose entire rollouts yield incorrect answers suffer from gradient vanishing. Hence, sampling high-difficulty questions with correct reasoning paths is crucial for LLM training. We first show that merely increasing rollout size does not consistently yield significant gains in *Pass@K* performance, and sometimes can even be harmful. We then quantify GRPO's cumulative advantage and highlight its under-weighting of high-difficulty samples.

**Naive Scaling of Rollout Size Benefits *Pass@1*, But Not Necessarily *Pass@K*.** We present the training dynamics of *Pass@1* and *Pass@K* performance during the RLVR training process in Figure 2. Enlarging the rollout size allows the sampling of correct solutions to hard problems during training. We originally assumed this would benefit *Pass@K* performance; however, experimental results show that this is not always the case. We find that Qwen2.5-Math-7B can significantly benefit from an increased rollout size, whereas for Qwen2.5-Math-1.5B, naively scaling rollout size can even harm *Pass@K* performance.

**Cumulative Advantage Bias in GRPO Variants hinders the improvement of *Pass@K*.** In the GRPO framework, the advantage estimation is derived by normalizing binary rewards:

$$\hat{A}_i^{std} = \frac{r_i - u}{\sigma}, \quad \hat{A}_i^{nostd} = r_i - u, \quad (1)$$

where $r_i$ is the binary reward of $i_{th}$ rollout, $u$ is the mean value of the group rewards $u = \text{mean}(\{R_i\}_{i=1}^G)$ and $\sigma$ is the standard deviation of the group rewards $\sigma = \text{std}(\{R_i\}_{i=1}^G)$. In the case of binary rewards, $u$ also represents the accuracy of LLM rollouts. Dr. GRPO (Liu et al., 2025b) removes the standard-deviation term from the advantage computation to eliminate question-level difficulty bias, and demonstrates its

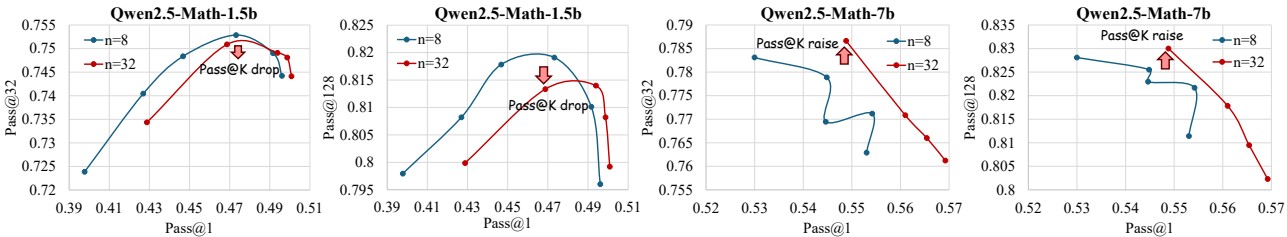

*Figure 2.* Training dynamics of *Pass@1* and *Pass@K* performance of Qwen2.5-Math-1.5B and -7B with different rollout size.

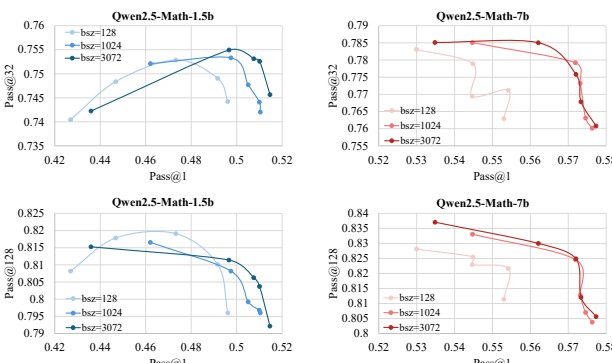

*Figure 3.* Training dynamics of *Pass@1* and *Pass@K* performance of Qwen2.5-Math-1.5B and -7B with different batch size.

superiority through extensive experiments. Consequently, the experiments reported in this study were conducted primarily though the Dr. GRPO methodology. We show more results of std-based advantage in Appendix E.2.

For a group $G$ with rollout size $N$, we define the **cumulative advantage** of a group as the sum of the absolute values of sample advantages: $\mathcal{A}_{\text{group}} = \sum_{i=1}^{G} |\hat{A}_i|$. The cumulative advantage reflects how much the algorithm weights each sample. Specifically, for Dr. GRPO,

$$\mathcal{A}_{\text{group}} = 2Nu(1 - u). \qquad (2)$$

The cumulative advantage functional curve is plotted in Figure 1. As shown in the figure, group-based advantage computation funnels its weight toward problems of medium difficulty while largely overlooking those that are highly difficult. This bias limits the *Pass@K* performance of RLVR.

### 2.2. Breadth: Iteration Instance Quantity in RLVR

We define **Breadth** as the number of instances used per iteration of the RLVR process. We'll show how increasing the batch size for the RLVR process improves the *Pass@1* performance.

**Breadth Matters for *Pass@1* Performance.** Most studies (Liu et al., 2025b;a; Yan et al., 2025; Fu et al., 2025) conventionally set the batch size to 128. In this subsection, we drastically increase the batch size to 3072 and plot the training dynamics of *Pass@1* and *Pass@32* performance in

Figure 3. Although large-batch optimization is not new in the broader deep learning literature, its effect in RLVR is notable: naively increasing the batch size brings a *Pass@1* improvement for all models, yet it harms the *Pass@128* performance of Qwen2.5-Math-1.5B. We consider that increasing the quantity of instances used in each iteration makes the gradient direction more accurate and reduces the impact of noise, thereby improving *Pass@1* performance.

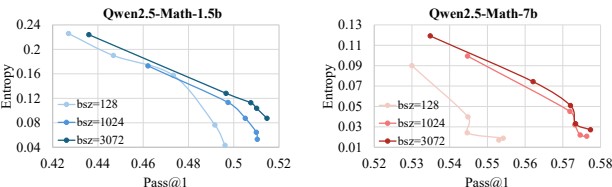

*Figure 4.* Training dynamics of *Pass@1* performance and token entropy for Qwen2.5-Math-1.5B and Qwen2.5-Math-7B.

### Breadth Sustains Entropy for Model Exploration.

High token entropy in LLMs indicates strong exploration capabilities. Our analysis shows a relationship between *Pass@1* and token entropy during training. As illustrated in Figure 4, increased training breadth enables LLMs to achieve higher entropy at a given Pass@1 accuracy. We consider a large training breadth acts as a form of entropy regularization, preventing premature convergence and boosting *Pass@1* performance while maintaining high entropy.

## 3. Methodology

In Section 2, we analyze the bias inherent in group-based advantage computation. To solve this issue, we introduce Difficulty Adaptive Rollout Sampling (**DARS**), which rebalances the cumulative advantage via multi-stage sampling. By combining this depth-oriented mechanism with a compatibility-driven large-breadth training scheme, we further obtain **DARS-Breadth**, which improves both *Pass@1* and *Pass@K*.

### 3.1. Difficulty Adaptive Rollout Sampling (DARS)

As shown in Figure 5, given a data batch $\mathcal{B} = \{q_j\}_{j=1}^{M}$ of reasoning questions, DARS operates in two phases: (i) **pre-rollout difficulty estimation** that assigns to each ques-

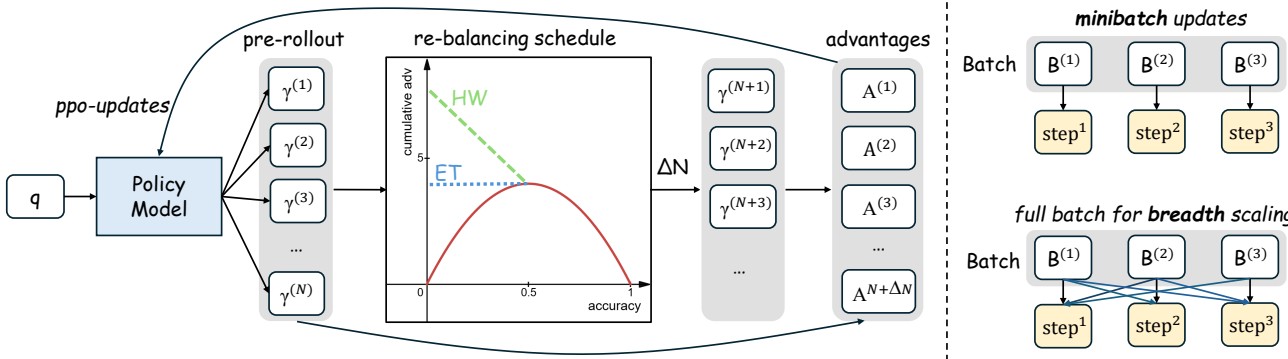

*Figure 5.* The overall training framework of our Difficulty Adaptive Rollout Sampling (**DARS**) with breadth scaling. Our DARS consists of 2 phases: 1) a pre-rollout stage to evaluate the difficulty of the given question, and 2) a re-balancing rollout stage to adjust the cumulative advantage. For breadth scaling, we replace the PPO minibatch with the full batch with multiple PPO epochs.

tion $q_j$ a scalar difficulty score $x_j \in [0, 1]$; and (ii) **multi-stage rollout re-balancing** that dynamically decides how many additional trajectories $\Delta n_j$ shall be allocated to $q_j$ so that the cumulative advantage for low-accuracy problems is up-weighted. To simplify the subsequent formula representation, we define

$$\mathcal{S}(\hat{a}_j) = 2\hat{a}_j(1 - \hat{a}_j). \tag{3}$$

**Phase 1: Pre-Rollout Difficulty Estimation.** For every $q_j$, we draw a light first-stage rollout consisting of $N^{pre}$ independent trajectories $\{\tau_j^{(i)}\}$. Let the per-trajectory reward be binary, $r_j^{(i)} \in \{0, 1\}$. We define the empirical accuracy

$$\hat{a}_j = \frac{1}{N^{pre}} \sum_{i=1}^{N^{pre}} r_j^{(i)}. \tag{4}$$

The difficulty score is then set to the complementary accuracy $x_j = 1 - \hat{a}_j$, so that $x_j \approx 1$ for the hardest problems and $x_j \approx 0$ for the easiest ones.

**Phase 2: Multi-Stage Rollout Re-Balancing.** Let $\mathcal{A}_{\text{group}}^N(u)$ denote the cumulative advantage under GRPO for a group whose average accuracy is $u$ with rollout size $N$. We aim to reallocate $\Delta N$ additional trajectories across the mini-batch so that the **effective** cumulative advantage for each question becomes an increasing function of its difficulty. To control the computing cost, we cap the rollout sampling upper limit at $N^{\max}$. To this end, we design two rebalancing schedules.

**Schedule 1: Equal-Treatment (ET).** For every question $q_j$ we enforce the rebalanced cumulative advantage as:

$$\mathcal{A}_{\text{group}}^{ET}(q_j) = \mathcal{A}_{\text{group}}^{N^{pre}}(0.5). \tag{5}$$

We raise the cumulative advantage of all difficulty problems ($\hat{a}_j < 0.5$) to the level achieved by a medium-difficulty

*Table 1.* Prompt-level objectives induced by different rollout allocation rules, where $p = \text{pass@1}(q)$ and $\mathbb{E}[\mathbf{g}_q]$ denotes the expected gradient contribution of prompt $q$. $C_{ET} = \frac{N^{pre}}{4}$ and $C_{HW} = \frac{N^{pre}}{2}$ are constant values.

| Method | $N(p)$ | $\mathbb{E}[\mathbf{g}_q]$ | Training Objective |
|---|---|---|---|
| **Dr. GRPO** | Fixed $N$ | $N \cdot \nabla_\theta p$ | **Pass@1** |
| **DARS-ET** | $\frac{N^{pre}}{4p(1-p)}$ | $C_{ET} \cdot \nabla_\theta \log \frac{p}{1-p}$ | **Log-Odds** |
| **DARS-HW** | $\frac{N^{pre}}{2p}$ | $C_{HW} \cdot \nabla_\theta \log p$ | **Max Likelihood** |

problem with accuracy $\hat{a}_j = 0.5$. The required extra trajectories are

$$\Delta n_j^{ET} = \min\left(\left\lceil \frac{\mathcal{A}_{\text{group}}^{N^{pre}}(0.5) - \mathcal{A}_{\text{group}}^{N^{pre}}(\hat{a}_j)}{\mathcal{S}(\hat{a}_j)} \right\rceil, N^{\max} - N^{pre}\right). \tag{6}$$

**Schedule 2: Hardness-Weighted (HW).** We now impose a monotonically increasing re-weighting that allocates more rollouts to lower-accuracy problems:

$$\mathcal{A}_{\text{group}}^{HW}(q_j) = 2x_j \mathcal{A}_{\text{group}}^{N^{pre}}(0.5). \tag{7}$$

This yields

$$\Delta n_j^{HW} = \min\left(\left\lceil \frac{2x_j \cdot \mathcal{A}_{\text{group}}^{N^{pre}}(0.5) - \mathcal{A}_{\text{group}}^{N^{pre}}(\hat{a}_j)}{\mathcal{S}(\hat{a}_j)} \right\rceil, \right. \tag{8}$$
$$\left. N^{\max} - N^{pre}\right).$$

**Implicit Prompt-Level Objectives of ET/HW Schedules.** Beyond rebalancing cumulative advantage, the ET/HW schedules also induce distinct *prompt-level* optimization targets. Up to constant scaling, Dr. GRPO optimizes **Pass@1**,

DARS-ET optimizes the **Log-Odds** of success, and DARS-HW optimizes the **Maximum-Likelihood** objective $\log p$, where $p = \text{pass@1}(q)$ denotes the prompt-level success rate. Table 1 summarizes the corresponding adaptive rollout budget, expected prompt-level gradient, and induced objective. The full derivation is deferred to Appendix B.2.

### 3.2. Depth Synergy with Breadth Scaling

Our analysis in Section 2.2 empirically confirms the substantial *Pass@1* improvements from large-breadth training. While DARS primarily optimizes training depth via multistage rollout rebalance, its dynamic batch-size adjustments preclude standard PPO-style mini-batch updates. To resolve this architectural constraint while leveraging breadth benefits, we replace PPO's mini-batch updates with full-batch gradient descent across multiple PPO epochs, as illustrated in Figure 5. This modification ensures compatibility with DARS's dynamic allocation while maximizing effective training breadth per optimization step. We term this integrated approach **DARS-Breadth**, unifying depth-adaptive sampling with breadth maximization.

Full-batch training offers two key advantages: (1) elimination of mini-batch gradient noise, and (2) sustained token-level exploration, acting as implicit regularization against premature convergence. Empirically, DARS improves *Pass@K* through depth optimization, while large-breadth training enhances *Pass@1*, highlighting their synergistic roles in RLVR.

### 3.3. Training Target

We adopt the clipped objective of GRPO without the KL penalty term. Following Dr. GRPO, we likewise remove the response length handling from the GRPO target. Specifically, for a problem $q$ sampled in training data $\mathcal{D}$, the training target is formalized as:

$$\mathcal{J}(\theta) = \mathbb{E}_{(q\sim\mathcal{D},\{o_i\}_{i=1}^{\mathcal{G}}\sim\pi_{\theta_{\text{old}}}(q)}\left[\frac{1}{G}\sum_{i=1}^{G}\sum_{t=1}^{|o_i|}\left(\right.\right.$$
$$\left.\left.\min\left(r_{i,t}(\theta)\hat{A}_{i,t},\ \text{clip}\left(r_{i,t}(\theta),1-\varepsilon,1+\varepsilon\right)\hat{A}_{i,t}\right)\right)\right], \quad (9)$$

where

$$r_{i,t}(\theta) = \frac{\pi_\theta(o_{i,t}\mid q,o_{i,<t})}{\pi_{\theta_{\text{old}}}(o_{i,t}\mid q,o_{i,<t})}. \quad (10)$$

The token advantage $\hat{A}_{i,t}$ is computed using Equation 1.

## 4. Experiments

### 4.1. Setup

**Evaluation and Training:** We evaluate the RLVR process using 5 widely used mathematical reasoning benchmarks: MATH-500 (Lightman et al., 2023), OlympiadBench (He et al., 2024), MinvervaMath (Lewkowycz et al., 2022), AIME24, and AMC23. We combine all of the evaluation benchmarks to report *Pass@1* (*Avg@128*) and *Pass@K* performance. The training data used in this work is OpenR1-45K, which is a subset of OpenR1-Math-220k (Hugging Face, 2025). Moreover, we adopt the same unbiased, low-variance estimator for *Pass@K* as used in prior works (Yue et al., 2025a; Chen et al., 2021a):

$$\text{pass@}K = \mathbb{E}_{x_i\sim\mathcal{D}}\left[1 - \frac{\binom{n-c_i}{k}}{\binom{n}{k}}\right].$$

Specifically, when $K = N$, the metric become: $\text{pass@}K = c_1 \vee c_2 \vee \cdots c_{128}$.

**Baseline and Methods:** We compare with: (1) *RLVR-baseline*: Dr. GRPO with rollout size 8 and batch size 128. (2) *Depth-Naive*: Simply increasing the rollout size to 32. (3) *Breadth-Naive*: Simply increasing the batch size to 3072. (4) *DARS-ET/HW*: Our algorithm introduced in Section 3.1 with Equal-Treat/Hardness-Weighted schedule, using batch size 128 and $N^{\max} = 32$. (5) *DARS-ET/HW-Breadth*: Our Depth-and-Breadth synergy algorithm introduced in Section 3.2, using batch size 3072 and $N^{\max} = 32$. For all methods, the number of PPO mini-steps is uniformly set to 2.

**Evaluation Protocol:** For all baselines, we select the checkpoint with the best *Pass@1* performance for reporting. For DARS, we selected the checkpoint that achieved the best *Pass@128* performance after surpassing the baseline *Pass@1* performance. Table 2 summarizes the *Avg@128* performance on each benchmark, the overall *Pass@1* across all test data, and the *Pass@128* performance.

### 4.2. Main Results

Breadth scaling delivers a clear and consistent boost to Pass@1. Across every model scale and every benchmark, Breadth-Naive outperforms both the RLVR baseline and Depth-Naive, lifting average Pass@1 (Avg@128) by 1.9–3.7 points on AIME24, MATH500, and Olympiad. This advantage is not merely additive: when depth is combined with breadth through DARS-Breadth, the margin widens further. DARS-Breadth reliably beats both Breadth-Naive and the DARS variants, confirming our central hypothesis: depth and breadth are complementary, not competing resources.

The practical impact is twofold. First, DARS-Breadth secures the highest Pass@1, the metric that matters most for single-shot deployment. Second, it matches the best

*Table 2.* Overall performance of *Pass@1* (*Avg@128*) and *Pass@128* of Qwen2.5-Math series.

| Method | AIME24 | MATH500 | Olympiad | AMC | Minerva | *Avg@128* | *Pass@128* |
|---|---|---|---|---|---|---|---|
| *Qwen2.5-Math-1.5B* | | | | | | | |
| Base Modle | 4.0 | 35.1 | 16.2 | 20.8 | 9.5 | 21.1 | 77.9 |
| RLVR baseline | 14.7 | 75.9 | 39.4 | 47.5 | 31.2 | 49.6 | 79.6 |
| Depth-Naive | 16.5 | 76.2 | 39.9 | 47.9 | 30.9 | 50.1 | 79.9 |
| Breadth-Naive | 18.5 | 77.6 | 41.7 | 49.8 | **31.9** | 51.5 | 79.2 |
| **DARS-1.5B-ET** | 15.8 | 76.0 | 40.9 | 47.2 | 30.0 | 50.0 | 81.2 |
| **DARS-1.5B-ET-Breadth** | 18.6 | **79.4** | **42.9** | 50.6 | 31.7 | **52.5** | 80.8 |
| **DARS-1.5B-HW** | 17.7 | 76.4 | 40.0 | 48.4 | 30.8 | 50.0 | 82.1 |
| **DARS-1.5B-HW-Breadth** | **19.3** | 79.0 | 42.7 | **51.9** | 31.6 | 52.4 | **82.2** |
| *Qwen2.5-Math-7B* | | | | | | | |
| Base Model | 11.6 | 52.3 | 19.7 | 35.2 | 15.3 | 30.1 | 82.1 |
| RLVR baseline | 26.8 | 82.2 | 44.3 | 57.2 | 35.7 | 55.3 | 81.4 |
| Depth-Naive | 28.0 | 83.8 | 46.4 | 59.0 | 37.3 | 57.0 | 80.3 |
| Breadth-Naive | 30.5 | 83.7 | 47.3 | 61.4 | 37.7 | 57.7 | 79.2 |
| **DARS-7B-ET** | 26.9 | 83.2 | 46.6 | 57.3 | **38.5** | 57.0 | 81.7 |
| **DARS-7B-ET-Breadth** | **33.3** | 83.8 | 47.8 | 61.3 | 38.4 | 58.1 | 82.1 |
| **DARS-7B-HW** | 30.1 | 83.5 | 47.1 | 59.4 | 37.2 | 57.3 | **83.5** |
| **DARS-7B-HW-Breadth** | 33.0 | **84.5** | **48.4** | **63.0** | 36.9 | **58.4** | 83.4 |
| *Llama-3.1-8B* | | | | | | | |
| Base Model | 0.23 | 6.13 | 1.54 | 2.76 | 2.72 | 3.25 | 52.7 |
| GRPO baseline | 0.66 | 29.6 | 7.09 | 10.1 | 15.7 | 15.8 | 56.5 |
| Depth-Naive | 0.43 | 33.6 | 9.40 | 12.3 | 19.7 | 18.9 | 58.6 |
| Breadth-Naive | 0.79 | 34.4 | 9.34 | 12.2 | 19.0 | 19.0 | 61.1 |
| **DARS-Llama-ET-Breadth** | **1.46** | **39.4** | **12.0** | 13.2 | **20.1** | **22.0** | 67.2 |
| **DARS-Llama-HW-Breadth** | 1.11 | 39.0 | **12.0** | **13.3** | 19.8 | 21.8 | **68.7** |

Pass@128 scores, demonstrating that the breadth-depth collaboration does not sacrifice the upper-bound capability revealed by heavy sampling. Finally, the choice of schedule matters: the HW schedule consistently yields superior Pass@K curves for both breadth and non-breadth training, while maintaining Pass@1 parity with the ET schedule, making it the preferred option across the board.

### 4.3. Training Dynamics

In this subsection, we further show more training dynamics to illustrate properties of existing RLVR methods and the superiority of our DARS and DARS-B.

***Pass@128* performance surpasses the base model, peaks quickly, and then declines.** We conduct RLVR experiments with rollout size 8/32 to compare our DARS (with $N^{\max} = 32$), the training dynamics of *Pass@128* performance during training is shown in Figure 6. Across all settings, *Pass@128* surpasses the base model during training, but declines after peaking, indicating that over-training

with RLVR harms *Pass@128* performance. Notably, DARS (with $N^{\max} = 32$) incurs substantially less inference cost than naively scaling the rollout size to n = 32. Despite this being an unfair comparison in terms of computational expenditure, our DARS attains the highest peak *Pass@128* performance and outperforms all other settings.

**Depth and Breadth Are Complementary in RLVR.** We show that Depth and Breadth are two complementary dimensions in RLVR, with breadth primarily governing *Pass@1* and depth primarily governing *Pass@K*. As shown in Figure 8, we present the *Pass@1–Pass@K* training-dynamics curves for the Breadth, Depth, and the two-dimensional synergy approach DARS-Breadth. The farther the *Pass@1–Pass@K* curve deviates outward, the more powerful the method. Our DARS-Breadth curves lie on the outermost envelope: it not only achieves the best *Pass@1*, but also simultaneously lifts *Pass@K*. This demonstrates the complementary roles of Depth and Breadth.

**DARS Elicits Longer Thought.** We tracked the response

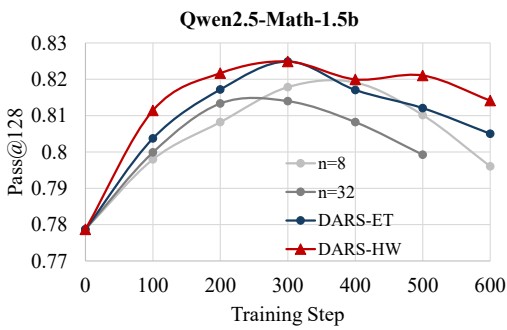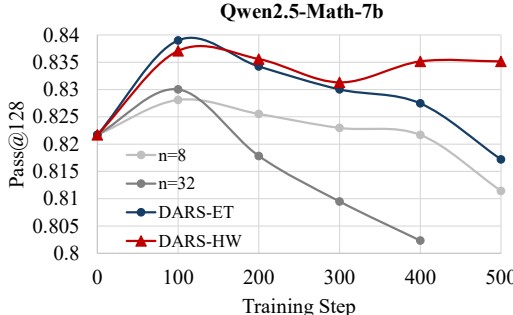

*Figure 6.* Training dynamics of *Pass@128* performance with different training steps of Qwen2.5-Math-1.5B and Qwen2.5-Math-7B.

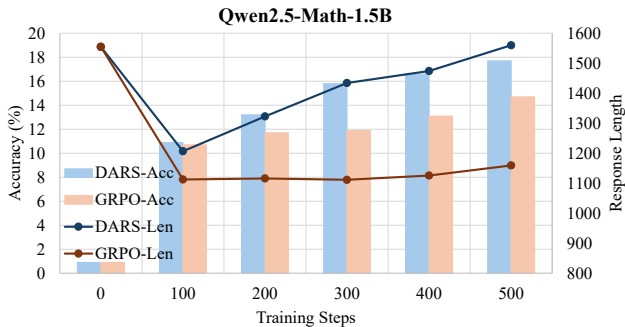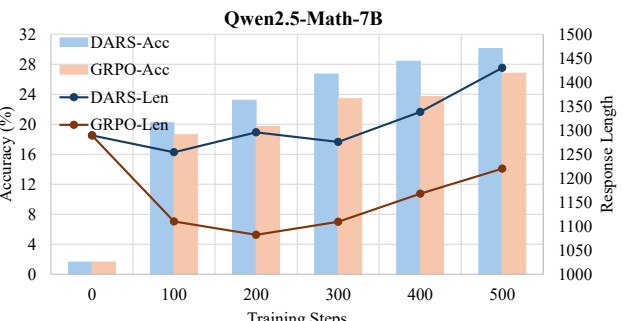

*Figure 7.* Comparison of our DARS-HW and Dr. GRPO Baseline on AIME 2024. We show the average response length and accuracy for Qwen2.5-Math-1.5B and Qwen2.5-Math-7B. After training with DARS, the models can achieve higher accuracy by utilizing longer thinking process.

length dynamics during the training of DARS and Dr.GRPO. As shown in Figure 7, on the challenging AIME 2024 benchmark, DARS consistently produces longer reasoning traces than the baseline and has higher accuracy. These results reveal that our DARS stimulates the model to perform deeper thinking to solve hard problems.

### 4.4. Test-Time Scaling Performance

We further evaluate whether DARS continues to benefit from additional test-time compute. Specifically, we compare the best checkpoints using majority voting (*maj@8*, *maj@16*) and *pass@128* on AIME24, AMC23, and MinervaMath. The *maj@k* results are averaged over five independent rounds of random sampling. We report this benchmark-level breakdown in Table 3, since the overall *pass@128* in Table 2 is aggregated across all evaluation instances and can partially mask gains on smaller but challenging benchmarks such as AIME24 and AMC23.

Table 3 shows that DARS consistently improves test-time scaling performance. For Qwen2.5-Math-1.5B, DARS-HW-Breadth improves the GRPO baseline by 4.6 points on AIME24 *maj@16*, 10.0 points on AIME24 *pass@128*, 2.7 points on AMC23 *maj@16*, and 2.2 points on Minerva-

Math *pass@128*. The gains are even larger on Qwen2.5-Math-7B, where DARS-HW-Breadth improves AIME24 *maj@8/maj@16* by 16.0/11.4 points, while also raising AMC23 *pass@128* and MinervaMath *pass@128* by 6.0 and 5.8 points, respectively. These results show that DARS not only improves single-sample behavior but also yields a stronger solution space under test-time search.

### 4.5. Ablation Studies

**Ablation Study on Base Model.** We illustrate the effectiveness of DARS on different models. The results are shown in Table 2. Our DARS-ET-Breadth achieves both the highest *Pass@1* and *Pass@128* performance, which further illustrates the effectiveness of our method. We further show the comparison results on static DARS in Appendix E.3.

**Ablation Study with openPangu.** We further evaluate DARS with openPangu-7B (Chen et al., 2025) on Huawei Ascend NPUs, to investigate the generalization ability of DARS on different models and hardware architectures. The experimental results are in Table 5 and DARS improves the performance of the openPangu model consistently on all datasets, more details are in Appendix E.8.

*Table 3.* Benchmark-level test-time scaling performance of the best checkpoints. *maj@k* is averaged over five independent sampling rounds.

| Method | AIME24 | | | AMC23 | | | MinervaMath | | |
|---|---|---|---|---|---|---|---|---|---|
| | *maj@8* | *maj@16* | *pass@128* | *maj@8* | *maj@16* | *pass@128* | *maj@8* | *maj@16* | *pass@128* |
| *Qwen2.5-Math-1.5B* | | | | | | | | | |
| RLVR Baseline | 19.3 | 22.7 | 56.7 | 53.7 | 56.8 | 91.6 | 33.9 | 35.4 | 58.8 |
| **DARS-HW** | 18.7 | 22.7 | 60.0 | 55.2 | 58.8 | **94.0** | **34.9** | **36.1** | 58.8 |
| **DARS-HW-Breadth** | **22.7** (↑3.4) | **27.3** (↑4.6) | **66.7** (↑10.0) | **58.6** (↑4.9) | **59.5** (↑2.7) | 93.9 (↑2.3) | 34.6 (↑0.7) | 35.4 | **61.0** (↑2.2) |
| *Qwen2.5-Math-7B* | | | | | | | | | |
| RLVR Baseline | 27.3 | 33.3 | 73.3 | 63.1 | 64.3 | 91.6 | 38.5 | **39.6** | 58.5 |
| **DARS-HW** | 37.3 | 43.3 | **76.7** | 68.0 | 69.6 | 95.2 | 38.6 | 39.3 | 62.1 |
| **DARS-HW-Breadth** | **43.3** (↑16.0) | **44.7** (↑11.4) | **76.7** (↑3.4) | **68.4** (↑5.3) | **69.9** (↑5.6) | **97.6** (↑6.0) | **39.0** (↑0.5) | 39.4 (↓0.2) | **64.3** (↑5.8) |

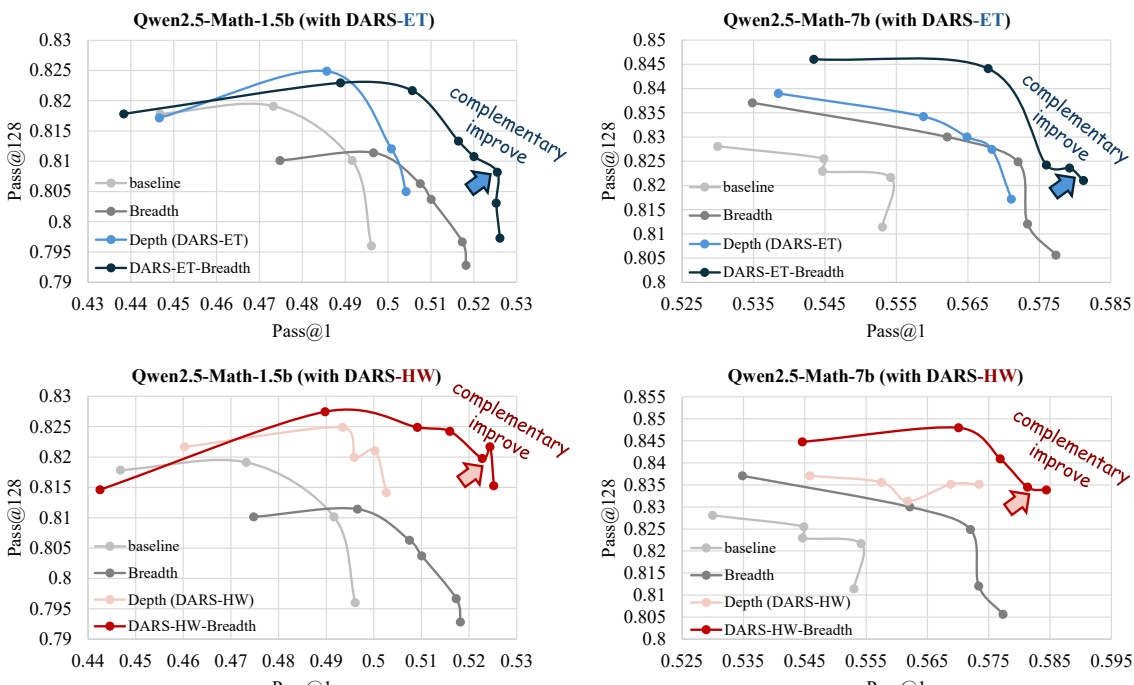

*Figure 8.* Depth and Breadth Synergy for *Pass@1* and *Pass@K* (K=128) performance.

## 4.6. Impact of Temperature

Some researches (Karan & Du, 2025; Qin et al., 2025; Ni et al., 2025) indicates that temperature matters in LLM reasoning. To further illustrate the performance improvement under different temperatures, we additionally add the above experiments. The results show that the improvement of our method is consistent over different temperatures. The results are shown in Figure 9.

## 4.7. DARS Improve *Pass@K* and Training Efficiency

Because the *Pass@K* (K=32/128) metric is hard to improve monotonically: it even starts to drop after prolonged training, while *Pass@1* remains comparatively stable and rarely collapses. We seek to boost *Pass@K* without degrading *Pass@1*. Figure 12 plots *Pass@128* against *Pass@1* under

a variety of experimental settings. It shows that, at any fixed *Pass@1* level, our DARS method delivers a consistently higher *Pass@128* than the other settings.

It is worth noting that, unlike the naive approach of simply increasing the rollout size to 32, our DARS achieves significantly higher training efficiency by allocating more rollouts to the hard problems. As shown in Table 4, our DARS methods need far fewer rollouts than the Depth-Naive method while achieving better performance.

## 4.8. Complete *Pass@K* Accuracy Curve.

We show the complete *Pass@K* curve for Llama-3.1-8B, Qwen2.5-Math-1.5B, and Qwen2.5-Math-7B in Figure 10. The 3 chosen models of DARS are: DARS-Llama-ET-Breadth, DARS-1.5B-HW-Breadth, and DARS-7B-HW-

| Qwen2.5-Math-1.5B | | | | | | Qwen2.5-Math-7B | | | | |
| --- | --- | --- | --- | --- | --- | --- | --- | --- | --- | --- |
| | *t=0.6* | *t=0.8* | *t=1.0* | *t=1.2* | *t=1.4* | | *t=0.6* | *t=0.8* | *t=1.0* | *t=1.2* | *t=1.4* |
| RLVR Baseline | 78.9 | 79.2 | 79.6 | 80 | 80 | RLVR Baseline | 80.4 | 80.7 | 81.4 | 81.6 | 81.6 |
| Depth-Naive | 79.4 | 79.6 | 79.9 | 80.3 | 80.4 | Depth-Naive | 79.7 | 80 | 80.3 | 80.7 | 80.8 |
| Breadth-Naive | 78.9 | 78.9 | 79.2 | 79.6 | 79.9 | Breadth-Naive | 77.7 | 78.1 | 79.2 | 79.4 | 79.6 |
| DARS-HW-Breadth | **81.6** | **81.9** | **82.2** | **82.6** | **82.7** | DARS-HW-Breadth | **82.6** | **82.8** | **83.4** | **84.0** | **84.0** |

*Figure 9.* Heat map of *Pass@128* of Qwen2.5-Math series in different temperatures.

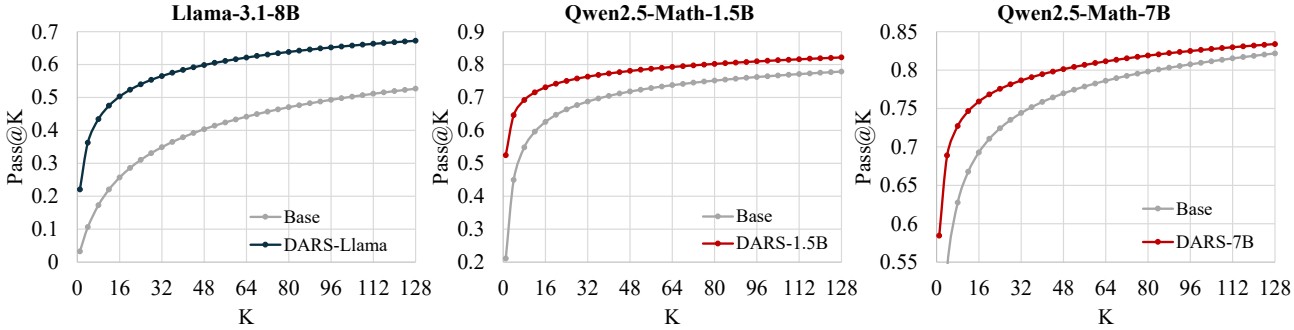

*Figure 10.* Complete *Pass@K* accuracy curve of base models and our DARS models.

*Table 4.* Average rollout numbers per prompt and wall-clock time per training step (seconds).

| Method | Avg. Rollouts / Prompt | Time / Step (s) |
| --- | --- | --- |
| *Qwen2.5-Math-1.5B* | | |
| Depth-Naive | 32 | 179 |
| DARS-ET | 15.2 (↓**52.5%**) | 115 (↓**35.8%**) |
| DARS-HW | 23.9 (↓**25.3%**) | 160 (↓**10.6%**) |
| *Qwen2.5-Math-7B* | | |
| Depth-Naive | 32 | 263 |
| DARS-ET | 12.8 (↓**60.0%**) | 174 (↓**33.8%**) |
| DARS-HW | 20.1 (↓**37.2%**) | 226 (↓**14.1%**) |

*Table 5.* DARS with openPangu on Ascend NPUs.

| Model | AIME24 | Math500 | Olympiad | AMC | Minerva |
| --- | --- | --- | --- | --- | --- |
| openPangu | 36.99 | 82.71 | 48.82 | 51.61 | 37.0 |
| **DARS** | **60.58** | **93.95** | **67.96** | **81.06** | **43.4** |

breakthrough of DeepSeek-R1 (Guo et al., 2025). Follow-up Large Reasoning Models like Kimi 1.5 (Team et al., 2025), Gemini-Think (DeepMind, 2024), QwQ (Qwen, 2024), and studies like (Zeng et al., 2025; Luo et al., 2025) further validate RLVR. The leading algorithm, GRPO (Shao et al., 2024), extends PPO (Schulman et al., 2017) with group-relative advantages, inspiring DAPO (Yu et al., 2025), VAPO (Yue et al., 2025b), and Dr. GRPO (Liu et al., 2025b). Yet GRPO and its variants systematically undervalue hard problems, hurting Pass@K. Inspired by Focal Loss (Lin et al., 2017), we propose DARS to explicitly up-weight the gradient contribution of challenging instances, ensuring the model explores sufficiently deep reasoning paths to unlock its potential. More related works are in Appendix A.

Breadth. DARS models demonstrate a breakthrough in the reasoning boundaries of the base model, especially on the LLama-3.1-8B model, where the improvement in Pass@k is particularly significant.

## 5. Related Works

Reinforcement Learning (RL) is now standard in post-training LLMs. After early reward-model pipelines (Ouyang et al., 2022a), Direct Preference Optimization (Rafailov et al., 2023) streamlined training by exploiting pairwise preferences. RL with verifiable rewards (RLVR) has since pushed reasoning benchmarks in math and code, culminating in OpenAI's o1 (Jaech et al., 2024) and the zero-RL

## 6. Conclusion

In this work, we reveal that GRPO-based RLVR methods under-weight hard problems due to cumulative-advantage bias, capping *Pass@K*. Our DARS sampler efficiently re-allocates rollouts to these hard instances, while large-breadth training with full-batch updates raises *Pass@1*. The unified DARS-Breadth jointly lifts *Pass@1* and *Pass@K*, proving depth and breadth are synergistic levers in RLVR.

## Acknowledgment

This work is partially supported by National Key R&D Program of China under Grant No. 2024YFA1012700, by the National Natural Science Foundation of China (NSFC) under Grant No. 62402410, by Guangdong Provincial Project (No. 2023QN10X025), and by HKUST(GZ) Kunpeng&Ascend Center of Cultivation. Zhijiang Guo is supported by the Guangdong Provincial Key Lab of Integrated Communication, Sensing and Computation for Ubiquitous Internet of Things (No. 2023B1212010007)

## Impact Statement

This paper presents work whose goal is to advance the field of Machine Learning. There are many potential societal consequences of our work, none which we feel must be specifically highlighted here.

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

# Appendix

# Contents of Appendix

## A. More Related Works

With the rapid advancement of RLVR and the proliferation of open-source LRMs, many studies have begun to analyze the RLVR pipeline and these open LRMs. Several studies (Liu et al., 2025a; Zhao et al., 2025; Shah et al., 2025) indicates that the self-reflect and self-critique behaviors observed after RLVR originates from the base model rather than the RL process. (Dang et al., 2025) find that although the RLVR process benefits *Pass@1*, *Pass@K* may decline as training progresses. Subsequently, (Yue et al., 2025a) through extensive experimental analysis, discovered that RLVR's performance is significantly constrained by the base model; once training converges, it struggles to surpass the capability boundary of the base model. These studies have sparked widespread concern about the capability ceiling of RLVR, and consequently, the *Pass@K* metric has become a focal point for diagnosing and potentially transcending the intrinsic limits imposed by the base model (Liang et al., 2025). This paper analyzes and refines the RLVR pipeline from the dual perspectives of *Pass@1* and *Pass@K*.

# B. Mathematical Derivations for DARS and GRPO Training Dynamics

## B.1. Derivation of Additional Rollouts $\Delta n_j$

The cumulative advantage for a group with accuracy $\hat{a}_j$ and total rollout size $N_j = N^{pre} + \Delta n_j$ is given by:

$$\mathcal{A}_{\text{group}}(\hat{a}_j, N_j) = N_j \cdot \mathcal{S}(\hat{a}_j),$$

where $\mathcal{S}(\hat{a}_j) = 2\hat{a}_j(1 - \hat{a}_j)$.

After the first-stage rollout of size $N^{pre}$, the initial cumulative advantage is:

$$\mathcal{A}_{\text{group}}^{N^{pre}}(\hat{a}_j) = N^{pre} \cdot \mathcal{S}(\hat{a}_j).$$

Our goal is to determine the number of additional trajectories $\Delta n_j$ needed so that the final cumulative advantage $\mathcal{A}_{\text{group}}(\hat{a}_j, N_j)$ meets a target value $\mathcal{A}_{\text{group}}^{\text{target}}(\hat{a}_j)$.

**Equal-Treatment (ET) Schedule:**

The target cumulative advantage is set to be constant for all questions with $\hat{a}_j < 0.5$:

$$\mathcal{A}_{\text{group}}^{\text{ET}}(\hat{a}_j) = \mathcal{A}_{\text{group}}^{N^{pre}}(0.5) = N^{pre} \cdot \mathcal{S}(0.5).$$

We solve for $\Delta n_j^{\text{ET}}$:

$$\mathcal{A}_{\text{group}}(\hat{a}_j, N_j) = \mathcal{A}_{\text{group}}^{\text{ET}}(\hat{a}_j)$$
$$(N^{pre} + \Delta n_j^{\text{ET}}) \cdot \mathcal{S}(\hat{a}_j) = N^{pre} \cdot \mathcal{S}(0.5)$$
$$\Delta n_j^{\text{ET}} \cdot \mathcal{S}(\hat{a}_j) = N^{pre} \cdot \mathcal{S}(0.5) - N^{pre} \cdot \mathcal{S}(\hat{a}_j)$$
$$\Delta n_j^{\text{ET}} = \frac{N^{pre} \cdot \mathcal{S}(0.5) - N^{pre} \cdot \mathcal{S}(\hat{a}_j)}{\mathcal{S}(\hat{a}_j)}.$$
$$\Delta n_j^{\text{ET}} = \frac{\mathcal{A}_{\text{group}}^{N^{pre}}(0.5) - \mathcal{A}_{\text{group}}^{N^{pre}}(\hat{a}_j)}{\mathcal{S}(\hat{a}_j)}.$$

The rollout size must be an integer, and we cap the total rollout sampling upper limit at $N^{max}$, so

$$\Delta n_j^{\text{ET}} = \min\left( \left\lceil \frac{\mathcal{A}_{\text{group}}^{N^{pre}}(0.5) - \mathcal{A}_{\text{group}}^{N^{pre}}(\hat{a}_j)}{\mathcal{S}(\hat{a}_j)} \right\rceil, N^{\max} - N^{pre} \right).$$

**Hardness-Weighted (HW) Schedule:**

The target cumulative advantage increases with difficulty:

$$\mathcal{A}_{\text{group}}^{\text{HW}}(\hat{a}_j) = 2(1 - \hat{a}_j) \cdot \mathcal{A}_{\text{group}}^{N}(0.5) = 2x_j \cdot N^{pre} \cdot \mathcal{S}(0.5).$$

We solve for $\Delta n_j^{\text{HW}}$:

$$\mathcal{A}_{\text{group}}(\hat{a}_j, N_j) = \mathcal{A}_{\text{group}}^{\text{HW}}(\hat{a}_j)$$
$$(N^{pre} + \Delta n_j^{\text{HW}}) \cdot \mathcal{S}(\hat{a}_j) = 2x_j \cdot N^{pre} \cdot \mathcal{S}(0.5)$$
$$\Delta n_j^{\text{HW}} \cdot \mathcal{S}(\hat{a}_j) = 2x_j \cdot N^{pre} \cdot \mathcal{S}(0.5) - N^{pre} \cdot \mathcal{S}(\hat{a}_j)$$
$$\Delta n_j^{\text{HW}} = \frac{2x_j \cdot N^{pre} \cdot \mathcal{S}(0.5) - N^{pre} \cdot \mathcal{S}(\hat{a}_j)}{\mathcal{S}(\hat{a}_j)}.$$

Again, using the baseline advantage notation $\mathcal{A}_{\text{group}}^{N^{pre}}(\hat{a}_j) = N^{pre} \cdot \mathcal{S}(\hat{a}_j)$, we obtain:

$$\Delta n_j^{\text{HW}} = \min\left( \left\lceil \frac{2x_j \cdot \mathcal{A}_{\text{group}}^{N^{pre}}(0.5) - \mathcal{A}_{\text{group}}^{N^{pre}}(\hat{a}_j)}{\mathcal{S}(\hat{a}_j)} \right\rceil, N^{\max} - N^{pre} \right).$$

## B.2. Implicit Optimization Objectives of the ET and HW Reallocation Rules

We next analyze the prompt-level population objectives induced by the reallocation schedules under the Dr. GRPO-style unnormalized group-relative advantage used in our main experiments. Let $p \in (0, 1)$ denote the true pass rate of a prompt $q$, and let the difficulty score satisfy $x = 1 - p$. In this subsection we use the idealized population approximation $u \mapsto p$, replace the empirical estimate $\hat{a}_j$ with its population value $p$, and ignore the ceiling operator and the cap $N^{\max}$. Under this approximation, define the surrogate prompt-level gradient

$$\mathbf{g}_q(N) \triangleq \sum_{i=1}^{N} (r_i - p) \nabla_\theta \log \pi_\theta(o_i|q), \tag{11}$$

where $r_i \in \{0, 1\}$ is the rollout reward. We condition on a fixed prompt $q$, assume the rollouts $o_1, \ldots, o_N$ are sampled i.i.d. from the current policy $\pi_\theta(\cdot|q)$, and write expectations with respect to this rollout randomness. This population surrogate keeps the derivation concise; using the exact sample-mean baseline $\bar{r}_N$ only changes constant factors by a finite-sample correction and does not alter the induced objective family.

> **Proposition B.2 (Idealized Population Objectives of ET and HW under Dr. GRPO)**
>
> In the idealized continuous analysis above, DARS-HW induces $\mathbb{E}[\mathbf{g}_q^{\mathrm{HW}}] = \frac{N^{pre}}{2} \nabla_\theta \log p$, which is proportional to the **Maximum-Likelihood** gradient. Likewise, DARS-ET induces $\mathbb{E}[\mathbf{g}_q^{\mathrm{ET}}] = \frac{N^{pre}}{4} \nabla_\theta \log \frac{p}{1-p}$, which is proportional to the **Log-Odds** gradient.

*Proof.* For a single rollout, the score-function identity gives

$$\begin{aligned}
\mathbb{E}[\mathbf{g}_q(N)] &= \sum_{i=1}^{N} \mathbb{E}\left[(r_i - p)\nabla_\theta \log \pi_\theta(o_i|q)\right] \\
&= \sum_{i=1}^{N} \left(\mathbb{E}\left[r_i \nabla_\theta \log \pi_\theta(o_i|q)\right] - p\,\mathbb{E}\left[\nabla_\theta \log \pi_\theta(o_i|q)\right]\right) \\
&= N \cdot \nabla_\theta p.
\end{aligned} \tag{12}$$

Under the same population approximation, the expected cumulative advantage is

$$\mathbb{E}[\mathcal{A}_{\mathrm{group}}(N, p)] = 2Np(1 - p). \tag{13}$$

The medium-difficulty reference generated by the first-stage pre-rollout is therefore

$$\mathcal{A}_{\mathrm{ref}} = 2N^{pre} \cdot (0.5)(1 - 0.5) = \frac{N^{pre}}{2}. \tag{14}$$

**For the HW schedule**, Equation 7 sets the target cumulative advantage to

$$\mathcal{A}_{\mathrm{target}}^{\mathrm{HW}}(p) = 2(1 - p)\mathcal{A}_{\mathrm{ref}} = (1 - p)N^{pre}. \tag{15}$$

Equating this target with the expected group cumulative advantage yields

$$2N_{\mathrm{HW}}(p)p(1 - p) = (1 - p)N^{pre} \quad \implies \quad N_{\mathrm{HW}}(p) = \frac{N^{pre}}{2p}. \tag{16}$$

Substituting this rollout allocation into the expected gradient gives

$$\mathbb{E}[\mathbf{g}_q^{\mathrm{HW}}] = N_{\mathrm{HW}}(p)\nabla_\theta p = \frac{N^{pre}}{2p}\nabla_\theta p = \frac{N^{pre}}{2}\nabla_\theta \log p. \tag{17}$$

So,

$$\mathbb{E}[\mathbf{g}_q^{\mathrm{HW}}] = C_{HW} \cdot \underbrace{\nabla_\theta \log p}_{\text{Maximum Likelihood}}, \tag{18}$$

where $C_{HW} = \frac{N^{pre}}{2}$ is a constant.

**For the ET schedule**, Equation 5 fixes the target cumulative advantage to the same medium-difficulty anchor:

$$\mathcal{A}_{\text{target}}^{\text{ET}}(p) = \mathcal{A}_{\text{ref}} = \frac{N^{pre}}{2}. \tag{19}$$

Equating this target with the expected group cumulative advantage yields

$$2N_{\text{ET}}(p)p(1-p) = \frac{N^{pre}}{2} \implies N_{\text{ET}}(p) = \frac{N^{pre}}{4p(1-p)}. \tag{20}$$

Therefore,

$$\mathbb{E}[\mathbf{g}_q^{\text{ET}}] = N_{\text{ET}}(p)\nabla_\theta p = \frac{N^{pre}}{4p(1-p)}\nabla_\theta p = \frac{N^{pre}}{4}\nabla_\theta \log \frac{p}{1-p}. \tag{21}$$

So,

$$\mathbb{E}[\mathbf{g}_q^{\text{ET}}] = C_{ET} \quad \underbrace{\cdot \nabla_\theta \log \frac{p}{1-p}}_{\text{Log-Odds}}, \tag{22}$$

where $C_{ET} = \frac{N^{pre}}{4}$ is a constant.

This proves the proposition. $\qquad\square$

Proposition B.2 reveals that the two schedules induce different population-level weighting functions under Dr. GRPO: ET equalizes cumulative advantage and yields a **Log-Odds** weighting, whereas HW scales compute linearly with hardness and yields a **Maximum-Likelihood** weighting. The implemented algorithm further enforces integer rollout counts, a rollout cap $N^{\max}$, and a two-stage pre-rollout floor. These implementation details discretize or truncate the idealized continuous objectives derived above, but do not change their functional form in this population analysis.

# C. Theoretical Connection Between DARS and Maximum Likelihood Reinforcement Learning

Recently, Maximum Likelihood Reinforcement Learning (MaxRL) (Tajwar et al., 2026) introduces a modified advantage function to approximate the Maximum Likelihood (ML) objective. Appendix B.2 already shows that the continuous uncapped HW schedule induces a gradient exactly proportional to $\nabla_\theta \log p$. In this section, we further provide a formal comparison to MaxRL and demonstrate that **our Difficulty-Adaptive Rollout Sampling (DARS) with the Hardness-Weighted (HW) schedule optimizes an expected objective identical to MaxRL**.

## C.1. Objective Equivalence via Cumulative Advantage

We first analyze the weighting behavior of both algorithms utilizing our proposed Group Cumulative Advantage ($\mathcal{A}_{group}$) framework. Let $N$ denote the number of rollouts for a given prompt $q$, and let $u \in (0, 1)$ denote the mean reward (i.e., empirical accuracy or pass rate), where the binary reward is $r_i \in \{0, 1\}$.

> **Proposition C.1 (Equivalence of Difficulty Weighting Profile)**
>
> Both MaxRL and DARS-HW impose a problem-level optimization weight that is strictly proportional to $(1 - u)$. Consequently, both algorithms share an identical inductive bias: linearly up-weighting the optimization focus on problems as their difficulty increases ($u \to 0$).

*Proof.* For MaxRL, the advantage is scaled as $\hat{A}_i^{MaxRL} = \frac{r_i - u}{u}$. For a group of $N$ rollouts, the expected number of successful trajectories ($r_i = 1$) is $Nu$, and the expected number of failed trajectories ($r_i = 0$) is $N(1 - u)$. The cumulative advantage is thus computed as:

$$\mathcal{A}_{group}^{MaxRL}(u) = \mathbb{E}\left[\sum_{i=1}^{N} \left|\hat{A}_i^{MaxRL}\right|\right]$$
$$= Nu \cdot \left|\frac{1 - u}{u}\right| + N(1 - u) \cdot \left|\frac{0 - u}{u}\right|$$
$$= N(1 - u) + N(1 - u) = 2N(1 - u).$$

For DARS-HW, we employ the unscaled advantage $\hat{A}_i = r_i - u$. Based on our proposed schedule (Equation 7), the target cumulative advantage is explicitly constrained to:

$$\mathcal{A}_{group}^{HW}(u) = 2(1 - u)\mathcal{A}_{group}^{N_{pre}}(0.5) = C \cdot (1 - u).$$

where $C = 2\mathcal{A}_{group}^{N_{pre}}(0.5)$ is a constant independent of the prompt difficulty $u$. Since $\mathcal{A}_{group}^{MaxRL}(u) \propto (1 - u)$ and $\mathcal{A}_{group}^{HW}(u) \propto (1 - u)$, the weighting profiles are equivalent. □

## C.2. Equivalence of Expected Gradients

To formalize the alignment in optimization trajectories, we compare the total expected gradient contribution of a single prompt $q$ under both methods. Let $\nabla J_i(u) = (r_i - u)\nabla_\theta \log \pi_\theta(o_i|q)$ denote the standard baseline-subtracted policy gradient for a single rollout.

> **Proposition C.2 (Equivalence of Expected Gradient Updates)**
>
> Let $K$ be a positive scalar. The expected gradient updates of MaxRL and DARS-HW are proportional: $\mathbb{E}[\nabla J_q^{MaxRL}] \propto \mathbb{E}[\nabla J_q^{DARS-HW}] \propto \frac{1}{u}\mathbb{E}[\nabla J_i(u)]$. Both methods approximate the Maximum Likelihood gradient by universally scaling the standard policy gradient by a factor of $1/u$.

*Proof.* For MaxRL, using a fixed rollout size $N$, the total expected gradient is scaled algebraically:

$$\mathbb{E}[\nabla J_q^{MaxRL}] = \mathbb{E}\left[\sum_{i=1}^{N} \frac{r_i - u}{u}\nabla_\theta \log \pi_\theta(o_i|q)\right] = \frac{N}{u} \cdot \mathbb{E}\left[\nabla J_i(u)\right].$$

For DARS-HW, we dynamically adjust the rollout size $N_{DARS}(u)$ to match the target $\mathcal{A}_{group}^{HW}(u) = C(1-u)$. Recall the standard group cumulative advantage property $\mathcal{A}_{group} = 2N_{DARS}u(1-u)$. Equating the two yields $2N_{DARS}(u)u(1-u) = C(1-u)$, which simplifies to $N_{DARS}(u) = \frac{C}{2u}$. Thus, the expected gradient contribution is:

$$\mathbb{E}[\nabla J_q^{DARS-HW}] = \mathbb{E}\left[\sum_{i=1}^{N_{DARS}(u)} (r_i - u)\nabla_\theta \log \pi_\theta(o_i|q)\right] = \frac{C}{2u} \cdot \mathbb{E}\left[\nabla J_i(u)\right].$$

Up to the constants $N$ and $C/2$, both expected gradients scale strictly by $1/u$. $\qquad\square$

## C.3. Variance Analysis: DARS-HW versus MaxRL

While Proposition C.2 establishes that DARS-HW and MaxRL optimize the same expected objective, they diverge significantly in their construction of the Monte Carlo gradient estimator. Let $\Sigma_g = \text{Var}(\nabla J_i(u))$ denote the trace of the covariance matrix of the unscaled token-level gradient for a single rollout. We make the standard assumption that conditioned on the prompt $q$ and current policy $\pi_\theta$, individual rollouts are independent and identically distributed (i.i.d.).

Let $u \to 0$ denote the regime of increasingly difficult reasoning tasks. For the MaxRL estimator, the variance relies on algebraic scalar multiplication over a fixed $N$ samples. By the properties of variance for i.i.d. variables ($\text{Var}(aX) = a^2\text{Var}(X)$):

$$\text{Var}(\hat{\nabla} J_q^{MaxRL}) = \text{Var}\left(\sum_{i=1}^{N} \frac{1}{u}\nabla J_i(u)\right) = \sum_{i=1}^{N} \frac{1}{u^2}\text{Var}(\nabla J_i(u)) = \frac{N}{u^2}\Sigma_g.$$

Conversely, DARS-HW physically expands the sampling space, drawing $N_{DARS}(u) = \frac{C}{2u}$ independent trajectories while maintaining the bounded unscaled advantage $|r_i - u| \le 1$:

$$\text{Var}(\hat{\nabla} J_q^{DARS-HW}) = \text{Var}\left(\sum_{i=1}^{N_{DARS}(u)} \nabla J_i(u)\right) = \sum_{i=1}^{N_{DARS}(u)} \text{Var}(\nabla J_i(u)) = N_{DARS}(u)\Sigma_g = \frac{C}{2u}\Sigma_g.$$

Taking the ratio of the two variances yields:

$$\frac{\text{Var}(\hat{\nabla} J_q^{MaxRL})}{\text{Var}(\hat{\nabla} J_q^{DARS-HW})} = \frac{N/u^2}{C/(2u)} = \frac{2N}{C} \cdot \frac{1}{u} \propto \frac{1}{u}.$$

Therefore, as $u \to 0$, the variance of MaxRL grows at a rate of $\mathcal{O}(1/u^2)$, while DARS-HW grows at a significantly slower rate of $\mathcal{O}(1/u)$.

## C.4. Theoretical Analysis Conclusions

- **Unified Optimization Objective with Divergent Implementation Routes.** DARS-HW and MaxRL are two distinct implementation routes toward the same Maximum Likelihood reinforcement learning objective. MaxRL achieves the target optimization weight by *algebraically modifying the advantage function* ($\hat{A}_i^{MaxRL} = \frac{r_i - u}{u}$), scaling the gradient signal of each rollout by $1/u$. In contrast, DARS-HW retains the original unscaled advantage function ($\hat{A}_i = r_i - u$) and instead achieves the identical expected gradient scaling by *adaptively increasing the number of rollouts* ($N_{DARS}(u) \propto 1/u$) for difficult problems.

- **Variance Reduction of DARS.** Appendix C.3 fundamentally explains the superiority of DARS over pure advantage-scaling methods as MaxRL. MaxRL heavily amplifies the gradient of rare, potentially idiosyncratic correct trajectories. This leads to a high-variance gradient regime, which may induce optimization instability and premature convergence. By contrast, DARS-HW functions as a variance-reduced estimator. Instead of amplifying a small fixed set of rollouts algebraically, DARS-HW allocates explicit computational budget to difficult problems, thereby expanding the breadth of **actual** correct reasoning paths discovered. This mechanism preserves a high signal-to-noise ratio in the gradient and maintains robust token-level entropy, theoretically validating DARS-HW as a more stable and efficient paradigm for Maximum Likelihood RL optimization.

# D. Training and Evaluation Details

> **Prompt for Solving Complex Reasoning Tasks**
>
> Your task is to solve the given question step by step. You should conduct a systematic, thorough reasoning process before providing the final answer. This involves analyzing, summarizing, exploring, reassessing, and refining your reasoning process through multiple iterations. Each reasoning step should include detailed analysis, brainstorming, verification, and refinement of ideas. You should include the final answer in \boxed{} for closed-form results like multiple choices or mathematical results.

**Parameters and Metrics.** Currently, our experiments are conducted with Qwen2.5-Math series language models (Yang et al., 2024). We set the temperature to 1.0 for both the training and evaluation procedures. In this paper, we mainly use two metrics, *Pass@1* and *Pass@K*. To acquire *Pass@K* results, we sample 128 candidate responses for each question during the evaluation process; the calculation of *Pass@1* is derived from *Avg@128*. Both the training and evaluation processes are scored using Math-Verify. The learning rate is 1e-6 for depth training methods, and 5e-6 for large breadth training. We do not use the reference model and KL loss. For fair comparison, we uniformly set the PPO mini step to 2 for all experiments. By default, the maximum prompt length is 1024, and the maximum response length is 3072 for the Qwen2.5-Math series model.

**Implementation Details.** Following LUFFY (Yan et al., 2025), we use the default subset and filter out generations that are longer than 8192 tokens and those that are verified wrong by Math-Verify [1], resulting in 45k question-solution pairs. For training Llama-3.1-8B, we use the train split of MATH dataset. Our training framework is derived from Verl (Sheng et al., 2024) pipeline, which is a flexible, high-performance reinforcement-learning framework built for training large language-model agents. With native PyTorch support and efficient distributed training, Verl lets researchers quickly prototype and scale RL algorithms like PPO on GPUs. Following Dr. GRPO (Liu et al., 2025b), we remove the KL loss and the length normalization in GRPO. For rollout generation, we use the modern serving stack provided by VERL with optimized backends such as vLLM. These backends employ continuous batching and asynchronous scheduling, which reduce padding waste and mitigate the straggler effect when different prompts receive different rollout budgets. All of our experiments are conducted on H200 GPUs. At present, the LLM of our experiment is the Qwen2.5-Math series.

**Training Steps and Checkpoint Steps.** For non-breadth methods on Qwen2.5-Math-1.5B/7B, we set the checkpoint step as 100. For breadth methods on Qwen2.5-Math-1.5B/7B, we set the checkpoint step as 15. The specific training steps are determined according to the convergence of the model. The number of training steps for non-breadth training is set as 300 for Llama-3.1-8B, 600 for Qwen2.5-Math-1.5B, and 500 for Qwen2.5-Math-7B. The number of training steps for breadth training is set to 70. For breadth training, we set the total training steps as 105 for Qwen2.5-Math-1.5B, and 75 for Qwen2.5-Math-7B.

---

[1] https://github.com/huggingface/Math-Verify

# E. More Experimental Results

## E.1. Generalization to Other Task Domains

To examine whether DARS is overly specialized to mathematical RLVR, we additionally evaluate the trained Qwen2.5-Math checkpoints on two out-of-domain benchmarks: GPQA-Diamond and HumanEval. These tasks differ substantially from the mathematical training domain and test scientific reasoning and code generation ability, respectively.

As shown in Table 6, DARS remains effective beyond mathematics. In particular, DARS-HW-Breadth consistently improves over the RLVR baseline (Dr. GRPO) on GPQA-Diamond (Rein et al., 2024) and HumanEval (Chen et al., 2021b) *Avg@64* for both model scales, while also improving HumanEval *Pass@64*. These results suggest that the core mechanism of DARS, allocating more sampling budget to difficult instances, is largely task-agnostic and can transfer to other reasoning domains.

*Table 6.* Additional out-of-domain evaluation on non-mathematical reasoning benchmarks.

| Method | GPQA-Diamond (*Avg@64*) | HumanEval (*Avg@64*) | HumanEval (*Pass@64*) |
|---|---|---|---|
| *Qwen2.5-Math-1.5B as the Base Model* | | | |
| RLVR Baseline | 28.5 | 41.1 | 84.2 |
| DARS-HW | 28.4 | 40.7 | **85.8** |
| **DARS-HW-Breadth** | **29.3** | **49.5** | 85.6 |
| *Qwen2.5-Math-7B as the Base Model* | | | |
| RLVR Baseline | 38.5 | 63.2 | 93.9 |
| DARS-HW | 38.6 | 63.7 | 94.8 |
| **DARS-HW-Breadth** | **39.6** | **68.3** | **94.9** |

## E.2. Ablation Study on std-based Advantage Computation

As illustrated in Section 2, Dr. GRPO (Liu et al., 2025b) removes the standard-deviation term from the advantage computation to eliminate question-level difficulty bias, and demonstrates its superiority through extensive experiments. Consequently, the experiments reported in this study were conducted primarily though the Dr. GRPO methodology. To further illustrate the effectiveness of DARS on std-based advantage computation, we conduct the experiment with HW schedule on Qwen2.5-Math-1.5B model, as shown in Figure 11.

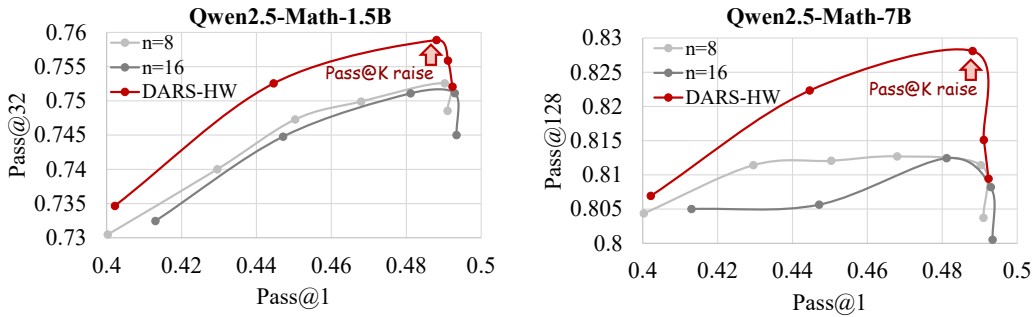

*Figure 11.* Comparison of our DARS on std-based advantage computation.

### E.3. Comparison with Static Difficulty Allocation

A natural heuristic baseline is to pre-compute a fixed difficulty score for each training problem using the initial base model, and then keep the rollout budget unchanged throughout RL training. We denote this variant as *Static DARS*. Specifically, we first evaluate the training set with Qwen2.5-Math-1.5B/7B before RL, estimate per-problem difficulty from the base-model accuracy, and assign fixed differentiated rollout budgets with the same HW schedule used by DARS-HW. The results are shown in Table 7. Static DARS consistently improves over the GRPO baseline and the naive uniform depth scaling baseline, showing that difficulty-aware budget allocation is useful. However, it still underperforms online DARS-HW on *Pass@128* for both model scales. This supports our claim that difficulty is policy-dependent and evolves during RL: once the model masters a subset of questions, a static allocation continues to spend rollout budget on them, whereas online DARS keeps refocusing compute on the current capability frontier.

*Table 7.* Comparison with heuristic baselines and a static difficulty-aware rollout allocation baseline. Static DARS computes difficulty scores once using the initial base model and keeps rollout budgets fixed during RL training.

| Method | AIME24 | Math500 | Olympiad | AMC | Minerva | Avg@128 | Pass@128 |
|---|---|---|---|---|---|---|---|
| *Qwen2.5-Math-1.5B as the Base Model* | | | | | | | |
| RLVR Baseline | 14.7 | 75.9 | 39.4 | 47.5 | 31.2 | 49.6 | 79.6 |
| Depth-Naive (N=32) | 16.5 | 76.2 | 39.9 | 47.9 | 30.9 | 50.1 | 79.9 |
| Static DARS | 17.3 | 76.7 | 40.3 | 47.3 | 31.2 | 50.3 | 80.4 |
| **DARS-HW** | 17.7 | 76.4 | 40.0 | 48.4 | 30.8 | 50.0 | **82.1** |
| *Qwen2.5-Math-7B as the Base Model* | | | | | | | |
| RLVR Baseline | 26.8 | 82.2 | 44.3 | 57.2 | 35.7 | 55.3 | 81.4 |
| Depth-Naive (N=32) | 28.0 | 83.8 | 46.4 | 59.0 | 37.3 | 57.0 | 80.3 |
| Static DARS | 28.6 | 84.0 | 46.7 | 59.7 | 37.6 | 57.4 | 81.1 |
| **DARS-HW** | 30.1 | 83.5 | 47.1 | 59.4 | 37.2 | 57.3 | **83.5** |

### E.4. Training Dynamics Comparison of DARS-ET/HW and Depth-Naive

We show this training dynamics comparison for Qwen2.5-Math-Series in Fig 12. The training dynamics illustrate the effectiveness of both DARS-ET and DARS-HW.

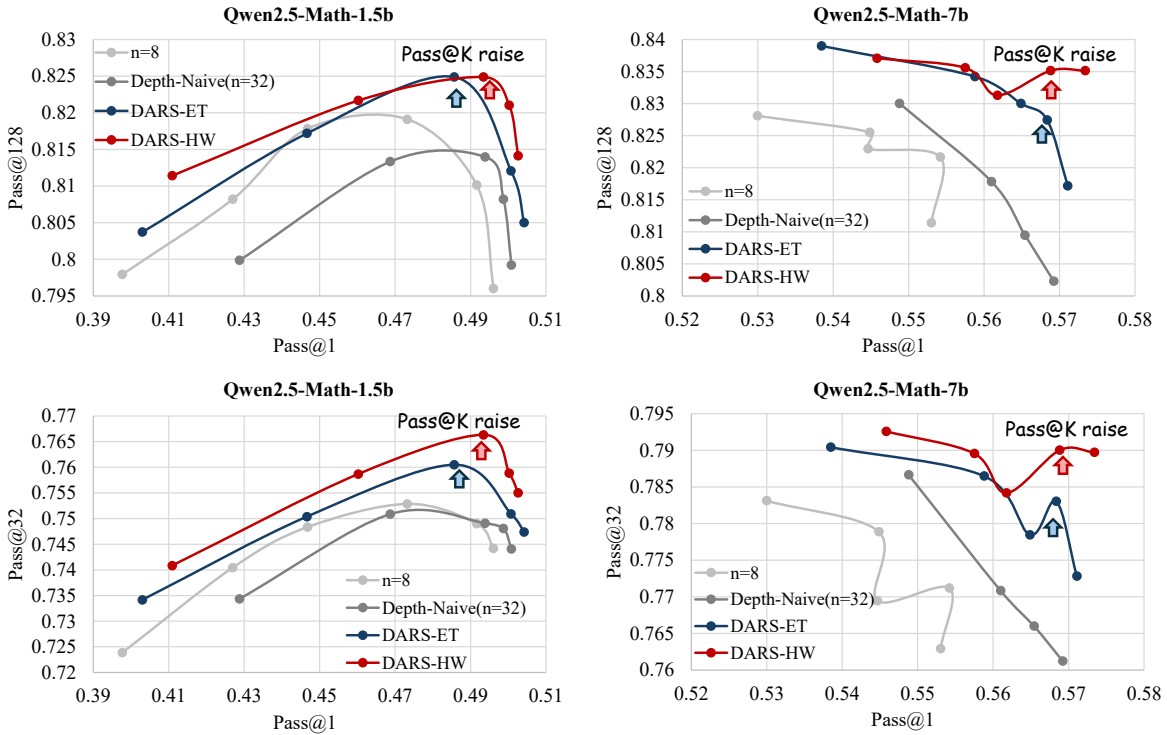

*Figure 12.* Training dynamics of *Pass@32/Pass@128* and *Pass@1* with different training steps of Qwen2.5-Math-1.5B and -7B.

### E.5. Depth and Breadth Synergy for Pass@1 and Pass@32

In Section 4.3, we show the training dynamics of *Pass@128-Pass@1* for DARS and baseline methods. To further illustrate the effectiveness of DARS, we show the training dynamics of *Pass@32-Pass@1* in Figure 14. Our DARS significantly improves the *Pass@32* performance compared to other methods.

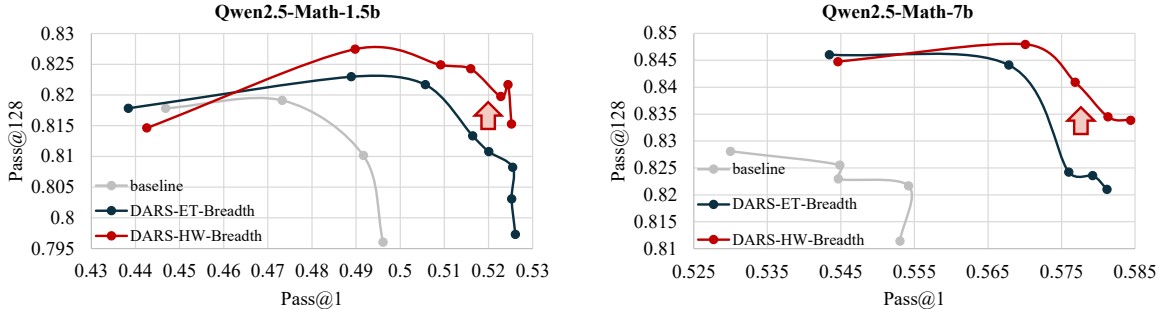

*Figure 13.* Comparison of ET and HW schedule in breadth training of Qwen2.5-Math series.

## E.6. Comparison of ET/HW Schedule in Breadth Scaling

In addition, compared with the ET schedule, DARS-HW-Breadth significantly improves the model's Pass@128 performance as shown in Figure 13. We consider this performance gain is due to the HW schedule placing greater emphasis on difficult samples.

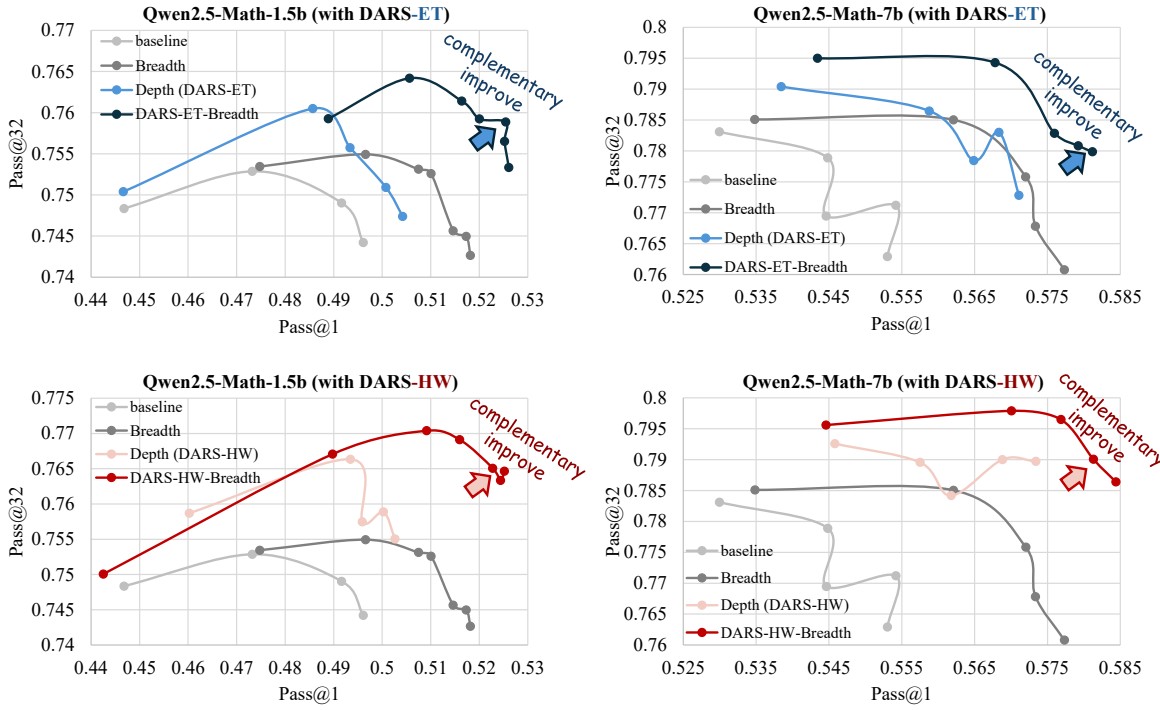

*Figure 14.* Complementary improve of Depth and Breadth Synergy for *Pass@1* and *Pass@K* (K=32) performance.

## E.7. Consistent Improvement During RL Process

To further show that our method consistently improve model performance, we calculated the mean of Pass@128 and Pass@32 for the last 3 checkpoints of each method, as shown in Table 8.

*Table 8.* Average performance of *Pass@1/32/128* for the last 3 checkpoints during training.

| Model | Pass@1 | Pass@32 | Pass@128 |
|---|---|---|---|
| *Qwen2.5-Math-1.5B as the Base Model* | | | |
| RLVR Baseline | 48.8 | 74.7 | 80.8 |
| Depth-Naive | 49.5 | 74.4 | 80.6 |
| Breadth-Naive | 51.4 | 74.4 | 79.8 |
| DARS-HW | 49.5 | 75.7 | 81.9 |
| **DARS-HW-Breadth** | **52.4** | **76.4** | **82.1** |
| *Qwen2.5-Math-7B as the Base Model* | | | |
| RLVR Baseline | 55.1 | 76.9 | 81.8 |
| Depth-Naive | 56.4 | 76.7 | 80.9 |
| Breadth-Naive | 57.2 | 76.7 | 81.3 |
| DARS-HW | 56.8 | 78.8 | 83.4 |
| **DARS-HW-Breadth** | **58.3** | **79.1** | **83.7** |

### E.8. Performance of Other Model

We further evaluate our method on Qwen2.5-7B-Instruct. Following (Liang et al., 2025), we change the training data to Math12k. The results are shown in Table 9.

To further demonstrate DARS can generalize across different models and hardware architectures, we also evaluate DARS with the openPangu-7B architecture on Huawei Ascend NPUs. We conduct our experiments on eight 910B4 NPUs (each of 32 GB memory), and report the Pass@32 performance in Table 10.

The above results show that our method outperforms the baseline in both the Pass@1 and Pass@K metrics.

*Table 9.* Overall performance of *Pass@1* (*Avg@128*) and *Pass@128* of Qwen2.5-7B-Instruct.

| Model | AIME24 | Math500 | Olympiad | AMC | Minerva | Avg@128 | Pass@128 |
|---|---|---|---|---|---|---|---|
| Qwen2.5-7B-Instruct | 11.9 | 72.3 | 37.1 | 42.2 | 31.9 | 47.2 | 80.3 |
| RLVR baseline | 14.2 | 74.8 | 37.6 | 43.4 | 33.4 | 48.6 | 78.8 |
| **DARS-HW-Breadth** | 15.6 | 76.5 | 38.4 | 44.7 | 34.6 | 49.6 | **82.3** |

*Table 10.* Overall performance of DARS with openPangu on Ascend NPUs.

| Model | AIME24 | Math500 | Olympiad | AMC | Minerva | Avg@32 | Pass@32 |
|---|---|---|---|---|---|---|---|
| openPangu-7B | 36.99 | 82.71 | 48.82 | 51.61 | 37.0 | 57.54 | 79.92 |
| **openPangu-7B + DARS** | **60.58** | **93.95** | **67.96** | **81.06** | **43.4** | **72.56** | **86.59** |

### E.9. Explanation of Significant Pass@k Gain for Llama-3

The stronger gains for Llama-3 likely stem from its base model characteristics. Recent studies (Gandhi et al., 2025; Wang et al., 2025) suggest that base models like Llama-3 lack mid-training, causing many valid responses to appear only at higher sampling budgets. As shown in the table below, Llama-3-8B-Base exhibits a much larger performance gap when increasing (K) from 8 to 32/64 compared to Qwen models as shown in Table 11. This indicates that Llama-3 has more correct solutions "hidden" deeper in the sampling space. Our DARS method, by focusing on hard examples, effectively uncovers these "deep" correct samples, leading to more substantial Pass@K improvements for Llama-3.

*Table 11.* Pass@K gap comparison of Qwen2.5-Math-1.5/7B and LLama3-8B-Base

| Model | Pass@8 | Pass@32 - Pass@8 | Pass@64 - Pass@8 |
|---|---|---|---|
| Qwen2.5-Math-1.5B | 56.5 | 12.4 | 17.1 |
| Qwen2.5-Math-7B | 63.5 | 10.5 | 14.4 |
| **Llama3-8B-Base** | 17.2 | **19.1** | **27.3** |

### E.10. Thinking Length Dynamics

This section investigates how DARS influences the reasoning length of LLMs. We tracked the response length dynamics during the training of Qwen2.5-Math-1.5B and 7B models. Our experiments reveal two key observations: (1) The training process shows a clear trend of increasing generation length, as shown in Figure 15. (2) When evaluated on AIME 2024, models trained with DARS consistently produce longer reasoning traces than the baseline, as shown in Figure 16. (For steps in multiples of 100, we take the average of 128 sampling results. To increase the density of data points, we also sample 16 times for every intermediate 50-step interval and take the average.) These results provide concrete evidence that our DARS method successfully stimulates the model to perform deeper and more thorough thinking.

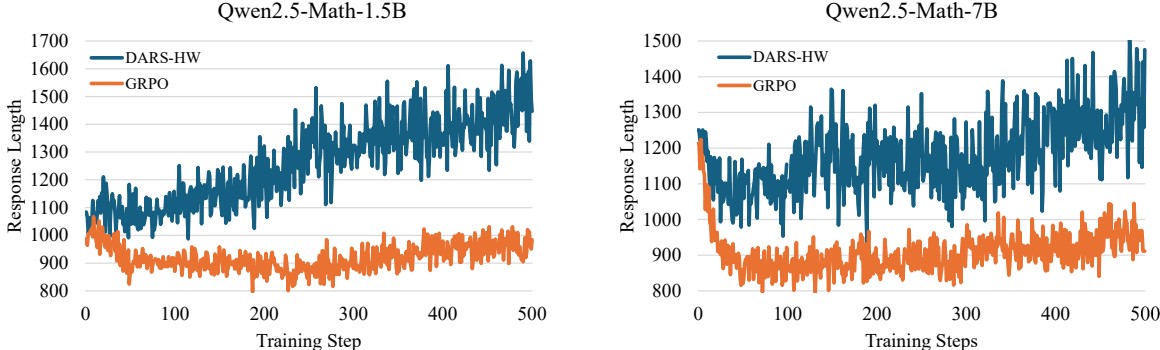

*Figure 15.* Training dynamics of response length for GRPO and DARS.

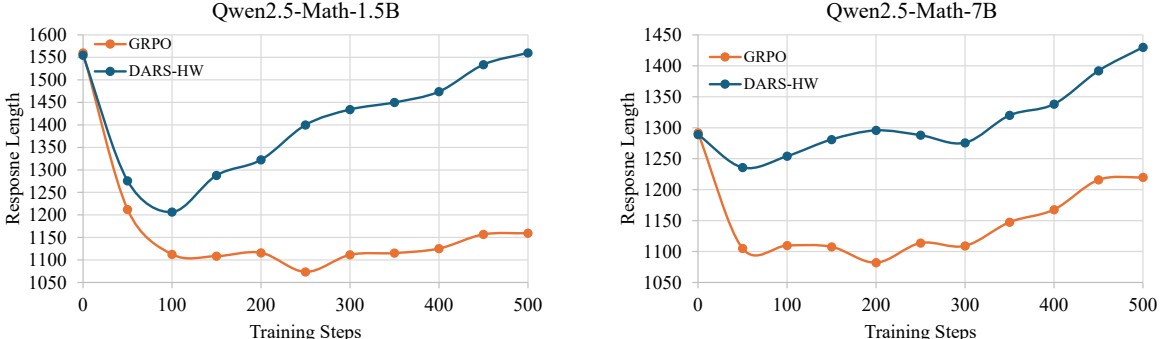

*Figure 16.* Statistical results of response length on AIME 2024 for GRPO and DARS.

## F. Case Study

We show an example of AIME24 to compare the difference for GRPO/DARS-HW trained model. As shown in Fig 17, compared to GRPO trained model, DARS-HW trained model will generate longer response to solve the hard question.

### Question

Let $ABC$ be a triangle inscribed in circle $\omega$. Let the tangents to $\omega$ at $B$ and $C$ intersect at point $D$, and let $\overline{AD}$ intersect $\omega$ at $P$. If $AB=5$, $BC=9$, and $AC=10$, $AP$ can be written as the form $\frac{m}{n}$, where $m$ and $n$ are relatively prime integers. Find $m + n$.

### Response of GRPO-Baseline

To solve the problem, we need to use some properties of the circle and the tangents. Here's the step-by-step solution:

1. **Identify the properties and relationships**:
   - Since $D$ is the intersection of the tangents to $\omega$ at $B$ and $C$, $D$ is the point where the tangents from $B$ and $C$ meet, making $DB$ and $DC$ tangent to the circle. This implies that $DB = DC$.
   - By the power of a point theorem, for point $D$, we have $DB^2 = DC^2 = DP \cdot DA$.

2. **Use the given side lengths**:
   - We are given $AB = 5$, $BC = 9$, and $AC = 10$. We need to find the length $AP$.

3. **Use the fact that $AD$ is a symmedian**:
   - Since $AD$ is the symmedian from $A$, we can use the property that the symmedian divides the opposite side in the ratio of the squares of the other two sides. So, $\frac{BD}{DC} = \frac{AB^2}{AC^2} = \frac{25}{100} = \frac{1}{4}$.

4. **Find the coordinates or use the properties of the triangle**:
   - We can use the fact that $D$ is the symmedian point and properties of the tangents to find the length $AP$.

5. **Use the formula for the length of the symmedian**:
   - The length of the symmedian can be found using the formula $AP = \frac{2 \cdot AB \cdot AC \cdot BC}{AB^2 + AC^2} \cdot \frac{1}{BC}$.

Let's calculate it step by step:

$$AP = \frac{2 \cdot 5 \cdot 10 \cdot 9}{5^2 + 10^2} = \frac{2 \cdot 5 \cdot 10 \cdot 9}{25 + 100} = \frac{900}{125} = \frac{36}{5}$$

So, $AP = \frac{36}{5}$. The values $m = 36$ and $n = 5$ are relatively prime, so $m + n = 36 + 5 = 41$.

The final answer is $\boxed{41}$. ✗

### Response of DARS-HW

To solve for $AP$ in the given triangle $ABC$ inscribed in circle $\omega$, we start by analyzing the given information and using properties of tangents and cyclic quadrilaterals.

Given:
- $AB = 5$
- $BC = 9$
- $AC = 10$

First, we use the fact that $D$ is the intersection of the tangents to $\omega$ at $B$ and $C$. This implies that $D$ is the excenter of $\triangle ABC$ opposite $A$. The tangents from $D$ to $\omega$ at $B$ and $C$ are equal in length, and $D$ lies on the line $AD$.

Next, we use the power of a point theorem at $D$. The power of point $D$ with respect to $\omega$ is given by:
$$DB^2 = DC^2 = DB \cdot DC$$

Since $D$ is the excenter, we can use the formula for the exradius $r_A$ of $\triangle ABC$ opposite $A$:
$$r_A = \frac{K}{s - a}$$
where $K$ is the area of $\triangle ABC$, and $s$ is the semi-perimeter of $\triangle ABC$.

First, we calculate the semi-perimeter $s$:
$$s = \frac{AB + BC + AC}{2} = \frac{5 + 9 + 10}{2} = 12$$

Next, we use Heron's formula to find the area $K$:
$$K = \sqrt{s(s-a)(s-b)(s-c)} = \sqrt{12(12-5)(12-9)(12-10)} = \sqrt{12 \cdot 7 \cdot 3 \cdot 2} = \sqrt{504} = 6\sqrt{14}$$

Now, we find the exradius $r_A$:
$$r_A = \frac{K}{s - a} = \frac{6\sqrt{14}}{12 - 5} = \frac{6\sqrt{14}}{7}$$

Since $D$ is the excenter, the tangents $DB$ and $DC$ are equal to $r_A$:
$$DB = DC = \frac{6\sqrt{14}}{7}$$

Now, we use the fact that $AD$ intersects $\omega$ at $P$. By the properties of the circle and the power of a point, we have:
$$AP \cdot AD = AB \cdot AC$$

We need to find $AD$. Using the formula for the length of the median from $A$ to $BC$ in $\triangle ABC$:
$$AD^2 = AB^2 + AC^2 - \frac{1}{2}BC^2 = 5^2 + 10^2 - \frac{1}{2} \cdot 9^2 = 25 + 100 - \frac{81}{2} = 125 - 40.5 = 84.5$$
$$AD = \sqrt{84.5} = \frac{\sqrt{169}}{2} = \frac{13}{2}$$

Now, we substitute $AD$ and the product $AB \cdot AC$ into the power of a point equation:
$$AP \cdot \frac{13}{2} = 5 \cdot 10 = 50$$
$$AP = \frac{50 \cdot 2}{13} = \frac{100}{13}$$

Thus, $AP$ can be written as $\frac{m}{n}$ where $m = 100$ and $n = 13$. Therefore, $m + n = 100 + 13 = 113$.

The final answer is:
$$\boxed{113}$$ ✓

*Figure 17.* A case study to demonstrate how DARS solves hard problems that the GRPO baseline misses. This is example is extracted from AIME24 for Qwen2.5-Math-7B.

# G. Discussion and Future Work

In this section, we analyze how hyperparameters $N$ and $N^{\max}$ control the shape of the cumulative advantage curve, and how this shape may influence training behavior. We further discuss how dynamically adjusting these parameters could enable a smooth transition from *Pass@K*-oriented to *Pass@1*-oriented training.

## G.1. Hyperparameter Control of Cumulative Advantage Shape

We show the Cumulative Advantage shape of ET/HW schedule with $N = 8$ in Figure 18. By continuously reducing the size of $N_{\max}$, the curve will contract accordingly. When $N_{\max} = N$, it is equivalent to the vanilla method without DARS.

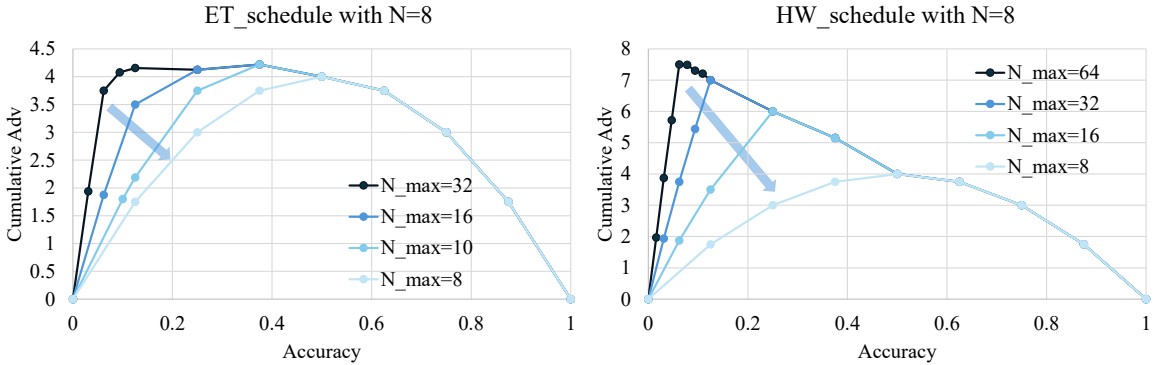

*Figure 18.* Control the shape of Cumulative Advantage by adjusting the $N_{max}$ hyperparameter of DARS.

## G.2. Potential Pass@K to Pass@1 Training Transition

The dynamic control of $N^{\max}$ suggests an intriguing training strategy: starting with a large $N^{\max}$ value to maximize *Pass@K* performance through extensive exploration of hard problems, then gradually reducing $N^{\max}$ throughout training to transition toward *Pass@1* optimization. This approach mirrors curriculum learning principles, where the training difficulty is progressively adjusted. Initially, the model benefits from the expanded solution space and diverse reasoning patterns discovered through heavy sampling on hard problems (high $N^{\max}$). As training progresses and the model's capability matures, reducing $N^{\max}$ focuses the training on refining the most promising solution strategies, ultimately improving single-shot performance.

To preliminarily validate this hypothesis, we conducted a new experiment on Qwen2.5-Math-1.5B with DARS-HW using an $N^{\max}$ annealing schedule. Specifically, we kept a high rollout cap in the early stage to emphasize exploration on hard problems, and then reduced the cap later in training to encourage convergence to the best solution paths. Concretely, we set $N^{\max} = 64$ for the first 200 training steps, and then linearly annealed it to 16 during the remaining training steps.

Table 12 shows that, compared with the fixed-budget DARS-HW setting ($N^{\max} = 64$ throughout training), the annealed variant slightly reduces *Pass@128* from 82.1 to 80.6, but improves *Pass@1* (*Avg@128*) from 50.0 to 50.9. This preliminary result is consistent with our hypothesis: a large early-stage $N^{\max}$ helps discover diverse valid trajectories, while a smaller late-stage $N^{\max}$ helps the policy concentrate on its best reasoning paths. Future work will explore more principled annealing schedules for $N^{\max}$ and investigate whether this transition strategy can simultaneously maximize both *Pass@1* and *Pass@K* performance, potentially overcoming the current limitations of RLVR training.

*Table 12.* Preliminary study of an $N^{\max}$ annealing schedule for transitioning from *Pass@K*-oriented exploration to *Pass@1*-oriented refinement.

| Method | Initial $N^{\max}$ | Final $N^{\max}$ | Pass@128 | Pass@1 (Avg@128) |
|---|---|---|---|---|
| DARS-HW (Fixed) | 64 | 64 | **82.1** | 50.0 |
| DARS-HW (Annealed) | 64 | 16 | 80.6 ($\downarrow$ 1.5) | **50.9** ($\uparrow$ 0.9) |

