# OpenReview forum: "Depth-Breadth Synergy in RLVR: Unlocking LLM Reasoning Gains with Adaptive Exploration"
_ICML.cc/2026/Conference — ICML 2026 regular_

### Official Review · Reviewer_sXaT · 2026-02-14

**Soundness:** 2
**Presentation:** 2
**Significance:** 2
**Originality:** 2
**Overall Recommendation:** 3
**Confidence:** 4

**Summary:**

This paper investigates two overlooked dimensions in the RLVR (Reinforcement Learning with Verifiable Rewards) training process: depth (problem difficulty) and breadth (training sample size). The authors discovered a hidden bias in the mainstream GRPO algorithm: it systematically underestimates the weight of high-difficulty problems through intra-group advantage calculation, making it difficult for the model to break through the Pass@K performance bottleneck. To address this, they proposed the DARS method, which dynamically allocates more rollouts to difficult problems through two-stage sampling; they also found that increasing the batch size for full-batch updates can significantly improve Pass@1. The most remarkable aspect is combining the two into DARS-Breadth, which simultaneously improves the performance of single inference and multiple samplings on multiple mathematical reasoning datasets, proving that depth and breadth are orthogonal and complementary optimization directions.

**Compliance With Llm Reviewing Policy:**

Affirmed.

**Final Justification:**

I will keep my initial rates.

**Key Questions For Authors:**

However, there are some aspects that raise some doubts.

1. For instance, although DARS has improved performance, the reasons for choosing either the HW or ET scheduling strategy are not fully explained. Figure 12 shows that HW performs significantly better in breadth training, but the paper does not delve into why the weighted strategy is more suitable for large batch scenarios.

2. In addition, the experiments mainly focused on mathematical reasoning and did not verify on tasks such as code generation that are also applicable to RLVR. This raises concerns about whether the method is limited to specific task types.

3. There is another minor detail. They mentioned that the Pass@K training would decline in the later stage (Figure 6), but they did not discuss the fundamental cause of this "overfitting" phenomenon, whether it was caused by the sparse rewards or the strategy collapse. If a more in-depth attribution analysis could be conducted by combining the changes in response length (Figure 10), it would be more convincing.

4. In the "Related Work" section, the responses to several recent studies that questioned the upper limit of RLVR capabilities (such as the "no 'aha moment'" statement made by Liu 2025a) were rather brief. In fact, a more positive discussion could be conducted on how DARS can alleviate these limitations.

**Limitations:**

See questions.

**Strengths And Weaknesses:**

The most remarkable aspect of this paper lies in its precise problem identification. It clearly depicts the phenomenon of "overemphasis on medium difficulty problems" in the GRPO using the mathematical form of cumulative advantage, and the visualization in Figure 1 is particularly intuitive. The design of DARS is also ingenious. It uses lightweight pre-sampling to estimate difficulty and then dynamically allocates resources, which not only solves the bias problem but also controls the computational cost. Table 2 shows that the actual average rollout number is actually 32 times less than naive while still being higher than it. The experimental part is quite solid. It not only verified on the Qwen series but also transferred to Llama and openPangu, even tested on the Ascend NPU. This indicates that the method has good generalization. What impresses me the most is that they did not just focus on Pass@K but also paid attention to Pass@1, revealing the "breadth" dimension that is often overlooked in RLVR. This dual perspective analysis should be very inspiring for subsequent work.

See questions for weakness.

---

> ### Author Rebuttal · Authors · 2026-03-29
>
> ## **General Response**
> We sincerely thank the reviewer for constructive feedback and insightful comments, and are encouraged that you recognize DARS’s performance improvements. We address your questions below based on your valuable suggestions.
>
> ## **1. Why HW Performs Better in Breadth Training (Question 1)**
> We appreciate you pointing out this nuanced empirical observation from Figure 12. The superiority of the Hardness-Weighted (HW) strategy in large-batch (breadth) scenarios stems from the synergy between **gradient stability** and **aggressive learning signals**.
>
> 1. **HW vs. ET Mechanism:** Equal-Treatment (ET) grants hard problems the *same* cumulative advantage as medium ones, while HW is far more aggressive: it assigns a *higher* cumulative advantage to the hardest problems, forcing the model to prioritize its capability frontier.
> 2. **Narrow Breadth Bottleneck:** Standard small-batch training has inherently high gradient variance. Hard problems have very low success rates, so their few sampled successful trajectories are often noisy or idiosyncratic. Over-weighting these signals in small batches exacerbates gradient instability, leaving HW performing similarly to or only marginally better than ET.
> 3. **Large Breadth Synergy:** Breadth training (large batches with full-batch updates) fundamentally smooths gradient noise. This stabilized optimization landscape lets the model safely digest HW’s aggressive learning signals from hard problems, providing the stable foundation for HW to fully unleash its potential and significantly outperform ET with large batches.
>
> ---
>
> ## **2. Generalization to Other Task Domains (Question 2)**
> We conducted additional experiments on OOD benchmarks. As shown below, DARS generalizes successfully to these new tasks, consistently outperforming the GRPO baseline. Its core mechanism: identifying and over-sampling hard instances, is fully task-agnostic, systematically finding correct logical paths for complex problems across domains.
>
> |Model|Method|GPQA-Diamond(avg@64)|HumanEval(avg@64)|HumanEval(pass@64)|
> |-|-|-|-|-|
> |Qwen2.5-Math-1.5B|GRPO-Baseline|28.5|41.1|84.2|
> ||DARS-HW|28.4|40.7|85.8|
> ||**DARS-HW-Breadth**|**29.3**|**49.5**|**85.6**|
> ||
> |Qwen2.5-Math-7B|GRPO-Baseline|38.5|63.2|93.9|
> ||DARS-HW|38.6|63.7|94.8|
> ||**DARS-HW-Breadth**|**39.6**|**68.3**|**94.9**|
>
> (GPQA-Diamond is a multi-choice task, thus we donot report pass@k performance)
>
> ---
>
> ## **3. Attribution Analysis of Pass@K Decline (Question 3)**
> This is a sharp observation. The late-training Pass@K decline (Figure 6) stems fundamentally from **strategy collapse (diversity loss)** driven by the RL objective, not sparse rewards.
>
> RLVR optimizes for *expected* reward (aligned with Pass@1). As training progresses, the policy maximizes this by converging on a single, highly reliable reasoning path per problem type.
> * **Link to Response Length (Figure 10):** The model learns to generate longer, meticulous reasoning chains to ensure correctness; once committed to this "safe but long" strategy, its token-level entropy drops (Figure 4).
> * **Impact on Pass@K:** Pass@K depends critically on exploration and diversity: if the first path fails, the model needs a *different* approach for subsequent samples. Strategy collapse leads the model to re-sample the same long trace; if this trace is flawed, all $K$ samples fail identically.
>
> Thus, Pass@K "overfitting" reflects the model over-optimizing for Pass@1 at the cost of generation diversity. DARS mitigates this early by forcing diverse path exploration for hard problems, but extended training eventually triggers collapse via PPO’s inherent mode-seeking nature.
>
> ---
>
> ## **4. Addressing the Upper Limit of RLVR and the "Aha Moment" (Question 4)**
> We fully agree our discussion of recent work (e.g., Liu 2025a) can be expanded to frame DARS more positively and forward-looking.
>
> Recent studies suggest RLVR does not organically create new knowledge (no "aha moment"), but elicits the base model’s pre-existing intrinsic capabilities. If this holds, RLVR’s "upper limit" is strictly defined by the hardest boundary problems the base model can solve with infinite sampling.
>
> However, **standard GRPO systematically fails to reach this upper limit**. Driven by the cumulative advantage bias we identified, GRPO ignores low-accuracy problems, leaving the base model’s deepest latent capabilities unelicited.
>
> **How DARS alleviates this:**
> DARS explicitly targets these boundary problems. By dynamically shifting compute ($\Delta n_j$) to the model’s most challenging problems, it forces the RL algorithm to learn from rare, serendipitous successes at the edge of its knowledge. While DARS still elicits the base model’s foundational knowledge, it maximizes the *depth* of this elicitation, significantly raising RLVR’s practical achievable upper limit and pushing the model far closer to its theoretical ceiling than standard RL pipelines.

---

> > ### Author Rebuttal · Reviewer_sXaT · 2026-04-03
> >
> > Thanks! I keep my initial rates.

---

> > > ### Author Response · Authors · 2026-04-03
> > >
> > > Dear Reviewer sXaT,
> > >
> > > Thank you very much for your acknowledgment of our rebuttal.
> > >
> > > We noticed that you updated the response status to "(b) Partially resolved - I have follow-up questions for the authors.", indicating that you still have some concerns. We take this feedback very seriously, as our goal is to ensure the utmost technical rigor and clarity of this work.
> > >
> > > However, since your latest comment ("Thanks! I keep my initial rates.") does not specify your question, we are currently at a loss as to how to best address your feedback.
> > >
> > > We would sincerely appreciate it if you could provide even a clarification on your follow-up question. We are fully committed to providing further evidence, additional experiments, or detailed explanations during this discussion period to resolve any lingering doubts you may have.
> > >
> > > We look forward to hearing from you and are ready to engage in further discussion.
> > >
> > > Best regards,
> > >
> > > Authors of Paper 6320

---

### Official Review · Reviewer_mpp8 · 2026-02-23

**Soundness:** 4
**Presentation:** 3
**Significance:** 4
**Originality:** 3
**Overall Recommendation:** 4
**Confidence:** 4

**Summary:**

This manuscript investigates the exploration bottlenecks in Reinforcement Learning with Verifiable Rewards, specifically focusing on the trade-offs between problem difficulty, referred to as Depth, and dataset coverage, referred to as Breadth. The authors identify a critical bias in standard group relative policy optimization algorithms, demonstrating that these methods inherently down-weight difficult, low-accuracy problems because the model rarely generates enough successful trajectories to form a meaningful advantage signal. To mitigate this, the authors propose Difficulty Adaptive Rollout Sampling, a dynamic mechanism that re-weights and heavily samples difficult problems early in training. They pair this with a curriculum-style annealing schedule that gradually reduces the maximum number of rollouts as training progresses, allowing the model to transition from deep exploration of hard problems to optimizing zero-shot inference performance.

**Compliance With Llm Reviewing Policy:**

Affirmed.

**Final Justification:**

I appreciate the authors' thorough rebuttal, which effectively clarified my technical doubts. While the manuscript still has room for improvement in terms of writing and figure presentation, I believe the core contribution and the newly provided clarifications now meet the threshold for ICML. Therefore, I am maintaining my Weak Accept score. I trust the authors will incorporate all the discussion points into the final version to ensure the paper's clarity and impact.

**Key Questions For Authors:**

1. How do you mathematically or empirically define the threshold where a problem is simply too difficult for the model, meaning that increasing the rollout budget will only result in compute waste rather than successful exploration?

2. Can you provide a more rigorous theoretical justification or a sensitivity analysis for your specific rollout annealing schedule, proving that it is not just an overfitted heuristic for your specific evaluation benchmarks?

3. How does the dynamic scaling of rollouts impact the overall wall-clock training time compared to a standard static baseline, and is the performance gain strictly proportional to the extra compute burned during the high-rollout exploratory phases?

Addressing the boundary conditions of compute waste on impossible problems would significantly strengthen my confidence in the real-world utility of this framework.

**Limitations:**

yes

**Strengths And Weaknesses:**

Overall, this paper tackles a highly practical and widely recognized bottleneck in the post-training of large language models. The observation that standard group relative policy optimization naturally ignores hard problems due to reward sparsity is both empirically sound and intuitively satisfying. By explicitly decoupling the exploration of complex reasoning traces from the optimization of simpler problems, the authors provide a compelling framework for scaling test-time compute during the training phase. The curriculum learning approach to annealing the rollout budget is a clever engineering solution that seemingly yields solid empirical gains.

However, the theoretical foundation and the robustness of the execution leave room for improvement, which grounds my recommendation as a Weak Accept. My primary concern is that the concepts of Depth and Breadth are treated somewhat colloquially and lack a rigorous mathematical formalization within the underlying Markov Decision Process. The proposed Difficulty Adaptive Rollout Sampling heavily relies on empirical heuristics rather than a principled algorithmic derivation.

Furthermore, aggressively scaling up the rollout budget for difficult problems introduces a massive risk of computational waste. If a base model lacks the intrinsic capability to solve a complex problem, generating sixty-four or even a hundred and twenty-eight rollouts will simply yield a batch of uniform failures, providing zero meaningful gradient signal while burning expensive accelerator time. The manuscript lacks a rigorous ablation study defining the boundary conditions where this adaptive sampling actually works versus where it merely wastes compute on insurmountable tasks.

Finally, relying on an explicit annealing schedule for the rollout budget introduces yet another highly sensitive hyperparameter trajectory into an already fragile reinforcement learning pipeline. While the authors show it works for their specific setup, practitioners adopting this method will likely face a daunting grid search to find the optimal decay rate for different model scales and dataset distributions.

---

> ### Author Rebuttal · Authors · 2026-03-29
>
> ## **1. Boundary Conditions & Compute Waste (Question 1 & Weakness 2)**
> We fully agree with your sharp observation. However, the fundamentally analogous to the static `group size` hyperparameter in GRPO baseline, which also lacks a strict mathematical boundary for optimality. The core goal of this work is to systematically analyze how to move beyond GRPO’s static heuristic via dynamic allocation of distinct group sizes to problems of varying difficulty.
>
> To directly address your concern about whether DARS enables effective exploration or merely burns compute, we tracked **Rollout Failure Rate** (defined as the share of rollouts from questions where the model failed to generate even one correct answer across all allocated attempts, i.e., absolute failure) on our training dataset. Average rates over the full training process are shown below:
>
> |Model|GRPO Baseline|DARS-ET|DARS-HW|
> |-|-|-|-|
> |Qwen2.5-Math-1.5B|28.7%|28.9%|**12.8%**|
> |Qwen2.5-Math-7B|17.2%|20.1%|**17.1%**|
>
> Rather than worsening compute waste, DARS-HW maintains or significantly reduces the overall rollout failure rate (15.9% drop for the 1.5B model). By intelligently allocating compute based on difficulty, DARS-HW uncovers correct reasoning traces for hard problems missed by the baseline, converting potential "waste" into high-value gradient signals.
>
> We further emphasize that problem difficulty in RLVR is policy-dependent and dynamic. To verify if early failures are permanently insurmountable, we tracked the number of completely failed questions during training. For DARS-HW with Qwen2.5-Math-1.5B, 31/128 samples had no correct answers at initiation; this number consistently fell to 9/128 by step 500. For Qwen2.5-Math-7B, failed samples dropped from 21/128 to 6/128. We further show this training dynamic in [[Anonymized Figure 2]](https://anonymous.4open.science/r/DARS-ICML-Rebuttal-8ED0/fig2.png).
>
> ---
> ## **2. Theoretical Justification for DARS (Question 2 & Weakness 1)**
> You may refer to `5. Justification for Reallocation Functions` of the response to Reviewer *Ueug*.
>
> ---
> ## **3. Annealing Schedule and Hyperparameter Sensitivity (Question 2 & Weakness 3)**
> We first respectfully clarify that **the annealing schedule is not the default setting of our DARS framework, nor was it used in the main experiments of our submitted paper.** In all our main tables and results, $N_{max}$ is kept strictly constant (e.g., $N_{max}=32$).
>
> The dynamic annealing of $N_{max}$ was strictly discussed in **Appendix F.2 as a promising future direction**. We apologize if its inclusion in the appendix caused any confusion regarding the fragility of our main pipeline. DARS works robustly out-of-the-box with a static $N_{max}$ without requiring a daunting grid search for decay rates.
>
> However, to address your question and further explore this future direction, we have conducted preliminary experiments on annealing $N_{max}$. **Our expectation is that starting with a high $N_{max}$ maximizes exploration (favoring Pass@K), while dynamically decaying it to a smaller value in later stages forces the model to converge on the best paths (favoring Pass@1).**
>
> As shown in the table below, our experimental results perfectly align with this expectation: the annealed strategy trades a slight drop in Pass@K for an improvement in Pass@1.
>
> |Method|Initial$N_{max}$|Final$N_{max}$|Pass@K(K=128)|Pass@1(Avg@128)|
> |-|-|-|-|-|
> |DARS-HW(Fixed)|64|64|82.1|50.0|
> |DARS-HW(Annealed)|64|16|80.6($\downarrow$1.5)|50.9($\uparrow$0.9)|
>
> ---
>
> ## **4. Wall-Clock Time vs. Static Baseline (Question 3)**
> To address your question regarding overall wall-clock time and whether the performance gain is proportional to the extra compute burned: **DARS actually reduces total compute compared to the standard scaling baseline while achieving better performance.**
>
> Instead of uniformly burning compute across all problems (like the Depth-Naive $N=32$ baseline), DARS *reallocates* compute from easy problems to hard problems. As a result, the average number of rollouts per prompt is significantly lower than the baseline, leading to strict wall-clock speedups per step.
>
> |Model|Method|Avg.Rollouts/Prompt|Time/Step(s)|Pass@128|
> |-|-|-|-|-|
> |**Qwen2.5-Math-1.5B**|Depth-Naive|32|179|79.9|
> ||**DARS-ET**|**15.2**($\downarrow$52.5%)|**115**($\downarrow$35.8%)|81.2|
> ||**DARS-HW**|**23.9**($\downarrow$25.3%)|**160**($\downarrow$10.6%)|**82.1**|
> ||||||
> |**Qwen2.5-Math-7B**|Depth-Naive|32|263|80.3|
> ||DARS-ET|12.8|174|81.7|
> ||**DARS-HW**|**20.1**($\downarrow$37.2%)|**226**($\downarrow$14.1%)|**83.5**|
>
> As shown above, the performance gain is **super-proportional** to the compute spent. DARS-HW on the 7B model achieves a +3.2% absolute gain in Pass@128 over the Depth-Naive baseline, while simultaneously consuming 37.2% *fewer* rollouts and reducing wall-clock time by 14.1%. By not wasting rollouts on problems the model already knows how to solve, DARS proves to be a highly efficient method.

---

> > ### Author Rebuttal · Reviewer_mpp8 · 2026-04-01
> >
> > I thank the authors for providing detailed experimental results that successfully address all of my initial questions. Consequently, I will maintain my positive rating.

---

> > > ### Author Response · Authors · 2026-04-03
> > >
> > > Dear Reviewer mpp8,
> > >
> > > Thank you for your time and efffort. We truly appreciate your constructive review and are happy to hear that the additional  detailed experimental results have addressed all of your initial questions.
> > >
> > > In light of this, we respectfully ask whether you might reconsider your overall recommendation or confidence rating to more accurately account for the enhanced clarity and technical soundness of our work. If you have other questions, we are willing to conduct further discussions.
> > >
> > > Best regards,
> > >
> > > The Authors 6320

---

### Official Review · Reviewer_fxUg · 2026-03-06

**Soundness:** 4
**Presentation:** 3
**Significance:** 4
**Originality:** 3
**Overall Recommendation:** 4
**Confidence:** 4

**Summary:**

This paper studies reinforcement learning with verifiable rewards (RLVR) for improving reasoning in large language models. The authors propose a framework that improves both exploration depth and breadth during training. The method introduces difficulty-adaptive rollout sampling (DARS) to prioritize harder problems and combines it with breadth scaling to increase training diversity. Experiments on reasoning benchmarks show improvements over standard RLVR baselines.

**Compliance With Llm Reviewing Policy:**

Affirmed.

**Final Justification:**

I appreciate the authors’ efforts during the rebuttal process. Overall, most of my concerns have been addressed, and I will maintain my positive score.

**Key Questions For Authors:**

- could you please provide some practical case analysis to show that how your method solve the problems?

- The proposed transition between Pass@K and Pass@1 optimization is an interesting direction. However, it remains unclear whether the two objectives can be simultaneously optimized in practice. Further analysis of the potential trade-off between Pass@K exploration and Pass@1 exploitation would strengthen the discussion.

- The experimental results demonstrate that the proposed method is effective in balancing Pass@1 and Pass@K during training. However, the approach relies on additional RL training, which can be computationally expensive. It would be interesting to discuss whether a similar trade-off between Pass@1 and Pass@K could be achieved at test time, potentially reducing the need for costly training.

**Limitations:**

same as last one

**Strengths And Weaknesses:**

Soundness: The approach is technically reasonable and supported by experiments comparing several RLVR baselines.

Presentation: The paper is generally clear and easy to follow.

Significance.
Improving RL training for reasoning models is an important problem. The idea of improving exploration in RLVR training is relevant for future reasoning models.

Originality.
The paper introduces a perspective that jointly considers exploration depth and breadth in RLVR training. The novelty mainly lies in the combination of adaptive sampling and scaling strategies.

---

> ### Author Rebuttal · Authors · 2026-03-29
>
> ### **General Response**
> We sincerely thank the reviewer for the constructive feedback and insightful questions. Your comments have highlighted excellent angles for further strengthening our analysis. Based on your suggestions, we address your specific questions below.
>
> ---
>
> ### **1. Practical Case Analysis (Question 1)**
> We completely agree that qualitative examples are essential to demonstrate *how* DARS solves hard problems that the baseline misses. As quantitatively shown in Section 4.3 and Appendix E.8 of our paper, DARS systematically elicits longer, more thorough thinking processes.
>
> To make this concrete, we will add a dedicated Case Study section in the revised Appendix. A representative example from the AIME 2024 is shown in [[Anonymized Figure 1]](https://anonymous.4open.science/r/DARS-ICML-Rebuttal-8ED0/fig1.png).
>
> ---
>
> ## **2. Simultaneous Optimization of Pass@1 and Pass@K (Question 2)**
> We first clarify that our combined method, **DARS-Breadth**, largely achieves simultaneous optimization. As shown in Figure 8 of the original manuscript, the training dynamics curve of DARS-Breadth wraps around the baseline and depth-only curves, pushing the Pareto frontier outward for both metrics simultaneously. Breadth (large batch size) acts as an implicit entropy regularizer sustaining token-level exploration, while DARS explicitly mines the correct paths for hard problems.
>
> However, to fully answer your question about strict optimization transitions, we conducted a new experiment based on our discussion in Appendix: Potential Pass@K to Pass@1 Training Transition. We implemented an $N_{max}$ annealing schedule: starting with a high $N_{max}$ for heavy hard-problem exploration (favoring Pass@K) in the first 200 steps, then linearly decaying $N_{max}$ to a smaller value in later training stages to force convergence on optimal paths (favoring Pass@1).
>
> |Method|Initial$N_{max}$|Final$N_{max}$|Pass@K(K=128)|Pass@1(Avg@128)|
> |-|-|-|-|-|
> |DARS-HW(Fixed)|64|64|82.1|50.0|
> |DARS-HW(Annealed)|64|16|80.6($\downarrow$1.5)|50.9($\uparrow$0.9)|
>
> ---
>
> ## **3. Train-Time RL vs. Test-Time Compute (Question 3)**
> We fully agree that RLVR training introduces computational overhead vs. direct test-time scaling (TTS) on base models, which we address from two perspectives: training efficiency and test-time superiority.
>
> **First, DARS is highly efficient among RLVR training methods.** While extending exploration costs compute, our adaptive approach is far more efficient than naively scaling rollout sizes uniformly across all prompts. As shown in Table 2 of our manuscript, DARS significantly reduces the generation burden by strictly reallocating compute only to the hardest instances:
>
> |Model|NaiveDepth|DARS-ET|DARS-HW|
> |-|-|-|-|
> |**Qwen2.5-Math-1.5B**|32AvgRollouts|15.2($\downarrow$52.5%)|23.9($\downarrow$25.3%)|
> |**Qwen2.5-Math-7B**|32AvgRollouts|12.8($\downarrow$60.0%)|20.1($\downarrow$37.2%)|
>
> **Second, Train-Time RLVR and Test-Time Compute (TTS) are strictly complementary.** Enhancing a model's intrinsic reasoning capability and boundary during training directly amplifies its performance ceiling during test-time search. To demonstrate this, we evaluated the TTS performance of our trained checkpoints using majority voting (`maj@16`, `maj@32`, `maj@64`) and `pass@128` on the AIME24, AMC, and Minerva. (Note: `maj@k` is averaged over five rounds of random sampling. For clarity, the overall `pass@128` in our manuscript’s Table 1 is a sample-level aggregate across all combined benchmarks, so pass@K gains are diluted by the large number of instances in MATH500 and Olympiad. We break down performance by individual benchmark below.)
>
>
> |Model|Method|AIME24(maj@8)|AIME24(maj@16)|AIME24(pass@128)|AMC(maj@8)|AMC(maj@16)|AMC(pass@128)|Minerva(maj@8)|Minerva(maj@16)|Minerva(pass@128)|
> |-|-|-|-|-|-|-|-|-|-|-|
> |**Qwen2.5-Math-1.5B**|GRPOBaseline|19.3|22.7|56.7|53.7|56.8|91.6|33.9|35.4|58.8|
> ||**DARS-HW**|18.7|22.7|60.0|55.2|58.8|**94.0**|**34.9**|**36.1**|58.8|
> ||**DARS-HW-Breadth**|**22.7**($\uparrow$3.4)|**27.3**($\uparrow$4.6)|**66.7**($\uparrow$10.0)|**58.6**($\uparrow$4.9)|**59.5**($\uparrow$2.7)|93.9($\uparrow$2.3)|34.6($\uparrow$0.7)|35.4|**61.0**($\uparrow$2.2)|
> |||||||||
> |**Qwen2.5-Math-7B**|GRPOBaseline|27.3|33.3|73.3|63.1|64.3|91.6|38.5|**39.6**|58.5|
> ||**DARS-HW**|37.3|43.3|**76.7**|68.0|69.6|95.2|38.6|39.3|62.1|
> ||**DARS-HW-Breadth**|**43.3**($\uparrow$16.0)|**44.7**($\uparrow$11.4)|**76.7**($\uparrow$3.4)|**68.4**($\uparrow$5.3)|**69.9**($\uparrow$5.6)|**97.6**($\uparrow$6.0)|**39.0**($\uparrow$0.5)|39.4($\downarrow$0.2)|**64.3**($\uparrow$5.8)|
>
> The results fully demonstrate the superiority of DARS in TTS scenarios. While DARS incurs an upfront training investment, it fundamentally improves the policy’s exploration space quality. Thus, applying TTS to a DARS-trained model delivers significantly higher, more robust reasoning performance than applying the same test-time compute budget to a standard GRPO model

---

> > ### Author Rebuttal · Reviewer_fxUg · 2026-04-03
> >
> > Overall, most of my concerns have been addressed, and I will maintain my positive score.

---

> > > ### Author Response · Authors · 2026-04-03
> > >
> > > Dear Reviewer fxUg,
> > >
> > > Thank you very much for your time and for acknowledging that your concerns have been fully resolved. We truly appreciate your thoughtful feedback and are glad to hear that our rebuttal has addressed the key questions you raised.
> > >
> > > Given that all your concerns are now resolved, would you kindly consider revising your overall recommendation or confidence score to better reflect the improved clarity and technical validity of our work? We believe that with these clarifications, the paper may be better positioned relative to your original assessment.
> > >
> > > Thank you again for your constructive review and valuable suggestions. If you have other questions, we are willing to conduct further discussions.
> > >
> > > Best regards,
> > >
> > > The Authors 6320

---

### Official Review · Reviewer_Ueug · 2026-03-12

**Soundness:** 2
**Presentation:** 3
**Significance:** 2
**Originality:** 2
**Overall Recommendation:** 3
**Confidence:** 4

**Summary:**

This paper investigates the exploration deficit in Reinforcement Learning with Verifiable Rewards (RLVR), specifically focusing on the GRPO algorithm, through the dual lenses of "Depth" (problem difficulty) and "Breadth" (number of training instances). The authors astutely identify that GRPO exhibits a systemic bias when calculating Cumulative Advantage, which disproportionately down-weights high-difficulty (low-accuracy) samples that are critical for advancing reasoning capabilities. To tackle the "Depth" issue, the authors propose Difficulty Adaptive Rollout Sampling (DARS), a multi-stage rollout mechanism that dynamically allocates more computational budget to harder problems, thereby reshaping the cumulative advantage distribution and significantly improving Pass@K performance. On the "Breadth" front, the authors demonstrate that simply scaling up the number of instances per iteration (e.g., using a batch size of 3072) with full-batch PPO updates effectively maintains token-level exploration entropy and reduces gradient noise, leading to substantial gains in Pass@1. Combining these insights, the proposed DARS-Breadth framework achieves robust improvements in both Pass@1 and Pass@K metrics across rigorous mathematical benchmarks (e.g., MATH500, AIME24).

**Compliance With Llm Reviewing Policy:**

Affirmed.

**Key Questions For Authors:**

1. System-Level Efficiency: In practical distributed training frameworks (e.g., Megatron or vLLM), how much computational idling (straggler effect/padding overhead) is introduced by the sequence length and generation count disparities inherent in DARS's multi-stage dynamic rollouts? Could you provide a direct wall-clock time comparison (not just rollout counts) against the baseline GRPO when reaching the same Pass@K performance?

2. Robustness of Pre-rollout: Given the extreme variance in reasoning path generation, is relying on a small $N^{pre}$ to determine $x_j$ sufficiently stable? If $\hat{a}_j$ is inaccurately estimated (e.g., the model gets a "lucky" correct answer on a hard prompt during pre-rollout), does this cause the model to permanently miss the opportunity to deeply explore that hard problem? Please provide a sensitivity analysis for the $N^{pre}$ parameter.
3. Heuristic Baselines: If you completely bypass online evaluation and instead use a static difficulty score (derived from a pre-trained reward model or prompt length heuristics) to assign a fixed, differentiated rollout count (Static DARS), how does the performance compare to your online DARS approach?

4. Justification for Equation (8): In the Hardness-Weighted (HW) strategy, the target advantage is set to $2(1-x_j)\mathcal{A}_{group}^{N^{pre}}(0.5)$. What is the theoretical derivation for this specific linear mapping? Have you tested alternative compute reallocation functions, such as softmax or exponential functions?

**Limitations:**

See Weaknesses.

**Strengths And Weaknesses:**

**Strengths:**

- **Insightful Problem Identification**: The mathematical analysis of Cumulative Advantage in GRPO is highly commendable. Precisely diagnosing the systemic underestimation of Group Relative Advantage for high-difficulty samples provides significant theoretical value for understanding the upper bounds of current RLVR paradigms.
- **Holistic Metric Optimization**: Differentiating the training dynamics of Pass@1 (driven by breadth) and Pass@K (driven by depth) offers a highly illuminating perspective. The empirical success of the combined DARS-Breadth framework strongly validates this dual-pronged approach.
- **Solid Empirical Evidence**: The ablation studies are comprehensive, demonstrating consistent results across models of varying parameter scales (Qwen2.5-Math-1.5B/7B, Llama-3.1-8B) and multiple challenging mathematical datasets.

**Weaknesses:**

- **Computational Inefficiencies and the Straggler Effect (System-Level)**: Although the paper claims that DARS requires fewer average rollouts than the naive Depth approach (N=32), it completely overlooks the severe "straggler effect" introduced by multi-stage dynamic sampling in parallel training systems. Within a single batch, simple prompts may require very few rollouts, while complex ones demand a massive generation budget. This extreme disparity in sequence lengths and generation counts inevitably leads to severe padding overhead and GPU idling. The absence of a wall-clock time and GPU utilization analysis critically undermines the engineering practicality of this method.
- **Triviality of "Breadth Scaling"**: Positioning "Breadth" as a core contribution is somewhat overstated, given that its implementation essentially boils down to scaling the batch size from 128 to 3072 and employing full-batch PPO updates without minibatches. In deep learning and RL, it is a foundational consensus that larger batch sizes provide more accurate gradient directions and reduce variance. Packaging this brute-force scaling—which imposes extreme demands on VRAM and multi-node communication bandwidth, as a novel "breadth mechanism" lacks algorithmic originality and is highly unfriendly to researchers with constrained compute resources.
- **High Variance in Pre-rollout Estimation**: The first stage of DARS relies on a very small number of trajectories ($N^{pre}$) to calculate the empirical accuracy $\hat{a}_j$ and assess difficulty. However, LLM sampling on complex reasoning tasks exhibits notoriously high variance. If $N^{pre}$ is small, the difficulty estimation $x_j$ will suffer from massive variance, leading to highly inaccurate rollout allocations ($\Delta N$) in subsequent stages. Conversely, increasing $N^{pre}$ contradicts the premise of a "lightweight assessment." The paper lacks a rigorous sensitivity analysis regarding the choice of $N^{pre}$ and its impact on the stability of the dynamic computational graph.

- **Lack of Advanced/Heuristic Baselines**: The core philosophy of DARS, allocating more compute to harder samples, which shares profound similarities with Prioritized Experience Replay (PER) or Curriculum Learning. The authors only compare their method against standard GRPO and naive scaling. A critical missing baseline is a "Static DARS" approach: what if the dataset was pre-scored for difficulty using an external reward model or offline heuristics (e.g., prompt length/complexity), and rollout budgets were statically assigned prior to training? The absence of such baselines weakens the argument that online, dynamic evaluation is strictly necessary.

---

> ### Author Rebuttal · Authors · 2026-03-29
>
> ## **1. System-Level Efficiency & The Straggler Effect**
> *   **First, modern RL frameworks inherently mitigate the straggler effect.** In `verl`, text generation is heavily optimized via **continuous batching** and **asynchronous concurrency**. When DARS combined with asynchronous execution that overlaps generation with other pipeline stages, padding overhead and GPU idling are fundamentally minimized at the system level.
> *   **Second, even without relying on asynchronous benefits, DARS is empirically much more efficient.** To provide the direct wall-clock comparison requested, we update Table 2 in the following to include the Time per Step (seconds).
>
> |Model|Method|Avg. Rollouts / Prompt|Time / Step (s)|
> |-|-|-|-|
> |**Qwen2.5-Math-1.5B**|Depth-Naive|32|179|
> || DARS-ET |15.2|115|
> || DARS-HW |23.9|160|
> |**Qwen2.5-Math-7B**|Depth-Naive|32|263|
> ||DARS-ET|12.8|174|
> ||DARS-HW|20.1|226|
>
> ---
>
> ## **2. Robustness of Pre-rollout & Sensitivity of $N_{pre}$**
> We conducted a sensitivity analysis on $N_{pre} \in \{4, 8, 12\}$ for Qwen2.5-Math-1.5B. The results indicate that while a smaller $N_{pre}$ (4) exhibits slightly lower performance due to higher estimation variance.
> Furthermore, as $N_{pre}$ increases from 8 to 12, the performance gains rapidly saturate. This validates our default choice of $N_{pre}=8$ as the "sweet spot" that successfully balances a lightweight difficulty assessment with statistical robustness.
>
> |Method|AIME|MATH500|Olympiad|AMC|Minerva|Avg@128|Pass@128|
> |-|-|-|-|-|-|-|-|
> |$N_{pre}=4$|17.2|75.8|39.7|47.3|30.4|49.6|81.3|
> |$N_{pre}=8$|17.7|76.4|40.0|48.4|30.8|50.0|82.1|
> |$N_{pre}=12$|16.9|76.2|39.9|48.6|31.3|49.9|82.2|
>
> ---
>
> ## **3. Triviality of "Breadth Scaling"**
> We respectfully clarify that we do not claim large batch sizes to be an algorithmically novel invention in the general deep learning context. Instead, our core contribution here is an empirical revelation specific to RLVR: Breadth (batch size) and Depth (difficulty scaling/rollouts) govern orthogonal reasoning capabilities (Pass@1 vs. Pass@K) and act synergistically. We will revise Section 2.2 to tone down the claims of algorithmic novelty regarding breadth and focus entirely on its synergistic role with Depth.
>
> ---
>
> ## **4. Heuristic Baselines / Static DARS**
> Following your advice, we evaluated the training dataset using the initial base models (Qwen2.5-Math-1.5/7B) to obtain static difficulty scores, and assigned fixed, differentiated rollout budgets prior to training. Static DARS outperforms the standard GRPO baseline, it falls behind our online DARS approach. The fundamental limitation of Static DARS is that **difficulty is policy-dependent and dynamic**.
>
> |Model|Method|AIME|MATH500|Olympiad|AMC|Minerva|**Avg@128**|**Pass@128**|
> |-|-|-|-|-|-|-|-|-|
> |Qwen2.5-Math-1.5B|GRPO Baseline|14.7|75.9|39.4|47.5|31.2|49.6|79.6|
> ||Depth-Naive|16.5|76.2|39.9|47.9|30.9|50.1|79.9|
> ||Static DARS|17.3 |76.7|40.3|47.3|31.2|50.3|*80.4*|
> ||**DARS-HW**| 17.7| 76.4| 40.0| 48.4| 30.8| 50.0| **82.1**|
> ||
> |Qwen2.5-Math-7B|GRPO Baseline|26.8|82.2|44.3|57.2|35.7|55.3|81.4|
> ||Depth-Naive| 28.0| 83.8| 46.4| 59.0| 37.3| 57.0| 80.3|
> ||Static DARS|28.6|84.0|46.7|59.7|37.6|57.4|*81.1*|
> ||**DARS-HW**|30.1|83.5|47.1|59.4|37.2|57.3|**83.5**|
>
> ---
>
> ## **5. Justification for Reallocation Functions**
> We sincerely appreciate this profound question regarding the theoretical underpinning of our design. We will next show through mathematical derivation that the specific linear mapping aligns to the policy gradient with principled statistical objectives: **Maximum Likelihood**  ($\nabla_\theta \log p$) and **Log-Odds**($\nabla_\theta \log \frac{p}{1-p}$).
>
>  1. Standard Expected Gradient: For a question $q$ with a group of $N$ rollouts, the expected policy gradient contribution is:
> $\mathbb{E}[g\_q] = N \cdot \nabla\_\theta p$
>
>  2. Cumulative Advantage: The expected Cumulative Advantage (the sum of absolute advantages) for a group is: $\mathbb{E}[{A}\_{group}(N)] = 2N \cdot p(1-p)$.
>
>  3. The Anchor: For a medium-difficulty problem ($p=0.5$) with the initial $N^{pre}$ rollouts, the reference advantage is: ${A}\_{r} = 2 N^{pre} \cdot (0.5)(1-0.5) = 0.5 N^{pre}$
>
>  4. The HW Mapping & Exact Rollout Size: For HW strategy, we have: $A^{HW}\_{target}(p) = 2(1-p) \cdot A\_r = (1-p) N^{pre}$.
> By equating the target to the expected advantage: $2N\_{HW} \cdot p(1-p) = (1-p) N^{pre}$, we have  $N\_{HW} = \frac{N^{pre}}{2p}$.
>
>     Recovering the **ML** Objective: Substituting this exact $N\_{HW}$ back into the expected gradient yields: $\mathbb{E}[g\_{HW}] = N\_{HW} \cdot \nabla\_\theta p = N^{pre}/2 \cdot \nabla_\theta \log p$
>
>  5. Similarly, applying the exact same derivation to ET schedule ($A^{ET}\_{target} = 0.5 N^{pre}$) yields $N\_{ET} = \frac{0.25 N^{pre}}{p(1-p)}$, which makes it an unbiased estimator for the **Log-Odds** objective ($\nabla\_\theta \log \frac{p}{1-p}$).
>
> (Due to rebuttal characters limitations, we will include the detailed derivation in the revised paper.)

---

> > ### Author Rebuttal · Reviewer_Ueug · 2026-04-03
> >
> > Thanks! I keep my initial rates.

---

> > > ### Author Response · Authors · 2026-04-03
> > >
> > > **Dear Reviewer Ueug,**
> > >
> > > Thank you very much for your acknowledgment of our rebuttal.
> > >
> > > We noticed that you updated the response status to “(c) Partially resolved or unresolved...”, indicating that there are still concerns regarding the core tenets of our paper that may require significant updates. We take this feedback very seriously, as our goal is to ensure the utmost technical rigor and clarity of this work.
> > >
> > > However, since **`your latest comment (“Thanks! I keep my initial rates.”) does not specify which particular concerns remain or which parts of our rebuttal were insufficient, we are currently at a loss as to how to best address your feedback.`**
> > >
> > > To further help clarify the theoretical foundations of our method, we have prepared the following summary table. It contrasts our two proposed variants (DARS-ET and DARS-HW) with the GRPO baseline in terms of expected gradient, exact rollout allocation $N(p)$, and the equivalent training objective.
> > >
> > >
> > > | Method | Expected Gradient $\mathbb{E}[\mathbf{g}_q]$ | Exact Rollout Allocation $N(p)$ | Training Objective |
> > > | :--- | :--- | :---: | :--- |
> > > | **Dr.GRPO** | $N \cdot \nabla_\theta p$ | Fixed $N$ | $\nabla_\theta \, \text{pass@}1(x)$ |
> > > | **DARS-ET (Ours)** | $\frac{0.25 N^{pre}}{p(1-p)} \nabla_\theta p$ | Adaptive: $\frac{0.25 N^{pre}}{p(1-p)}$ | $C_{\text{ET}} \cdot \nabla_\theta \log \frac{\text{pass@}1(x)}{1 - \text{pass@}1(x)}$  (Log-Odds) |
> > > | **DARS-HW (Ours)** | $\frac{0.5 N^{pre}}{p} \nabla_\theta p$ | Adaptive: $\frac{0.5 N^{pre}}{p}$ | $C_{\text{HW}} \cdot \nabla_\theta \log \text{pass@}1(x)$  (Maximum Likelihood) |
> > >
> > > where $C_{\text{ET}}$ and $C_{\text{HW}}$ are constant values.
> > >
> > > *(Note: We will expand the Appendix in the revised manuscript to include the complete theoretical derivation of these implicit maximum‑likelihood objectives from the reallocation functions.)*
> > >
> > > > We want to be completely transparent with you: we have invested a tremendous amount of time and effort into addressing every single concern raised by all reviewers. Resolving all outstanding issues has been and remains, is our utmost goal throughout this rebuttal process. We have carefully revisited our derivations, performed additional analyses (as summarized above), and revised the manuscript accordingly. If there is any remaining point that still falls short of your expectations, we are fully prepared to invest even further effort, conduct additional experiments if needed, or provide any other form of clarification, until you are fully satisfied.
> > >
> > > We would sincerely appreciate it if you could provide a brief clarification on the unresolved issues. We are fully committed to providing further evidence, additional experiments, or detailed explanations during this discussion period.
> > >
> > > **We look forward to hearing from you and are ready to engage in further discussion to solve all your concerns.**
> > >
> > > Best regards,
> > >
> > > Authors of Paper 6320

---

### Decision · Program_Chairs · 2026-04-30

**Decision:**

Accept (regular)

**Comment:**

This paper makes a genuinely insightful contribution to RLVR for LLM reasoning. Its most valuable aspect is the clear diagnostic analysis of GRPO’s cumulative advantage bias, supported by mathematical formulation and intuitive visualizations. The decoupling of depth (hard-problem exploration) and breadth (batch scaling) provides a novel and practically useful framework that moves beyond ad-hoc hyperparameter tuning. The rebuttal convincingly addressed the major technical concerns with rigorous additional experiments. While the breadth component is less novel, the synergy between depth and breadth is well demonstrated, and the overall empirical evidence is solid.

I recommend accept. The authors should incorporate the rebuttal’s additional analyses into the final manuscript to further strengthen the paper.